# MatFormer: Nested Transformer for Elastic Inference

**Devvrit**[*][△][◇]  **Sneha Kudugunta**[*][†][◇]  **Aditya Kusupati**[*][†][◇][+]

**Tim Dettmers**[†]  **Kaifeng Chen**[◇]  **Inderjit Dhillon**[◇][△]  **Yulia Tsvetkov**[†]  **Hannaneh Hajishirzi**[†]

**Sham Kakade**[‡]  **Ali Farhadi**[†]  **Prateek Jain**[◇][+]

[◇]Google Research  [△]University of Texas at Austin  [†]University of Washington  [‡]Harvard University

## Abstract

Transformer models are deployed in a wide range of settings, from multi-accelerator clusters to standalone mobile phones. The diverse inference constraints in these scenarios necessitate practitioners to train foundation models such as PaLM 2, Llama, & ViTs as a series of models of varying sizes. Due to significant training costs, only a select few model sizes are trained and supported, limiting more fine-grained control over relevant tradeoffs, including latency, cost, and accuracy. This work introduces MatFormer[2], a nested Transformer architecture designed to offer elasticity in a variety of deployment constraints. Each Feed Forward Network (FFN) block of a MatFormer model is jointly optimized with a few nested smaller FFN blocks. This training procedure allows for the Mix'n'Match of model granularities across layers – i.e., a trained universal MatFormer model enables extraction of *hundreds* of accurate smaller models, which were never explicitly optimized. We empirically demonstrate MatFormer's effectiveness across different model classes (decoders & encoders), modalities (language & vision), and scales (up to 2.6B parameters). We find that a 2.6B decoder-only MatFormer language model (MatLM) allows us to extract smaller models spanning from 1.5B to 2.6B, each exhibiting comparable validation loss and one-shot downstream evaluations to their independently trained counterparts. Furthermore, we observe that smaller encoders extracted from a universal MatFormer-based ViT (MatViT) encoder preserve the metric-space structure for adaptive large-scale retrieval. Finally, we showcase that speculative decoding with the accurate and *consistent* submodels extracted from MatFormer can further reduce inference latency.

## 1  Introduction

Large Foundation models (Anil et al., 2023; OpenAI, 2023; Dehghani et al., 2023) are deployed in a variety of settings like real-time response on mobile phones or in batch setting on multi-cluster GPUs for web-scale serving. To handle such varied settings, each model family provides a few *independently trained* models of different sizes. In order to cover a wide range of applications, typically these models' sizes are nearly linear on log-scale. For example, Llama family provides models with 7B, 13B, 33B and 65B parameters (Touvron et al., 2023a).

Such an approach has two key drawbacks: (a) as the models are independently trained, they incur significant overhead for colocation during inference and are not behaviorally consistent with

---

[*]Equal technical contribution. [+]Aditya Kusupati and Prateek Jain led the project.
  Correspondence: `devvrit@cs.utexas.edu, snehakudugunta@google.com,`
  `kusupati@cs.washington.edu, prajain@google.com`

[2]MatFormer stands for 🪆 **Mat**ryoshka Trans**former** due to the model's inherent nested nature.

Workshop on Advancing Neural Network Training at 37th Conference on Neural Information Processing Systems (WANT@NeurIPS 2023).

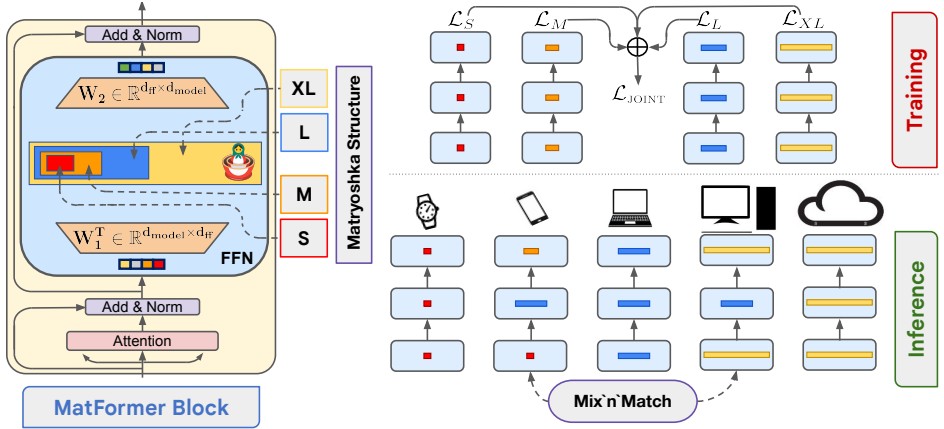

Figure 1: MatFormer introduces nested structure into the Transformer's FFN block & jointly trains all the submodels, enabling free extraction of hundreds of accurate submodels for elastic inference.

each other which are detrimental to inference optimization techniques like speculative decoding (Leviathan et al., 2023) and model cascades (Wang et al., 2020c), and (b) due to training overhead, practitioners typically train only a few models which do not cover the entire set of downstream use-cases. For example, a deployment setup might, say, have the latency budget to support 40B parameter Llama model, but can only host a 33B variant because the next bigger model (65B) has significantly higher latency. So, one would need to settle for a less accurate model despite the larger latency budget. While model compression approaches aim to address this issue, they typically require additional training for each model that needs to be extracted. Furthermore, when applied to LLMs, these techniques are known to significantly drop the accuracy (Jaiswal et al., 2023).

In this paper, we propose MatFormer, a natively elastic Transformer (Vaswani et al., 2023) architecture that allows for training one *universal* model which can be used to extract hundreds of smaller submodels without *any additional training* (Figure 1). MatFormer is a general architecture that can be applied to both encoders and decoders, is domain agnostic, and is compatible with most design choices and training pipelines of large Transformer-based models – LLMs & ViTs.

MatFormer follows the principle of matryoshka representation learning (Kusupati et al., 2022), to introduce nested substructure inside the standard Transformer block. Formally, MatFormer defines a Transformer blocks $T_i$, such that, $T_1 \subset T_2 \subset \cdots \subset T_g$, where $g$ is the number of nested transformer blocks, and $T_i \subset T_{i+1}$ relation indicates that the parameters of $T_i$ are contained in those of $T_{i+1}$. MatFormer can induce such sub-structure in both the attention and the feedforward network (FFN) blocks of the Transformer (see Figure 1). Consider a FFN block that has $d_{\mathrm{ff}}$ neurons in the hidden layer. Then, MatFormer induces matryoshka structure on these neurons, where $T_i$ contains the first $m_i$ neurons and $1 \le m_1 \le m_2 \cdots \le m_g = d_{\mathrm{ff}}$ represent the number of neurons for each granularity or sub-model. Intuitively, this implies that the first $m_1$ neurons are "most significant" neurons as they belong to all the blocks followed by the next $m_2 - m_1$, and so on. We can form a similar substructure on the attention heads, with the heads being organized from "most" to "least" significant, where the more significant heads are shared by more sub-models. That is, we use only the first $m_i$ attention heads for the $i$th granularity. In fact, we can also introduce this sub-structure in the token embedding ($d_{\mathrm{model}}$) supplied to each Transformer block.

However, in most LLMs and ViTs, the FFN block in the Transformer accounts for more than $60\%$ non-embedding parameters and is responsible for the largest chunk of latency during inference. So, in this work, we focus on inducing the MatFormer's nested sub-structure in the FFN block. We then stack the individual blocks (for $l$ layers) to form $g$ nested models ($\mathcal{M}_{1\cdots g}$) with shared parameters i.e., $\mathcal{M}_i \subset \mathcal{M}_{i+1}$. Finally, we jointly train these $g$ models by combining each model's loss.

This leads to a natural question: can one extract more than $g$ models after inducing the MatFormer structure? Yes, in fact, it is possible to extract exponentially many models. Using the trained MatFormer blocks $T_1, \ldots, T_g$ at each layer, one can form new models by Mix'n'Match, i.e., by taking an arbitrary combination of these blocks across layers. For example, in the first layer, one can select $T_g$, the largest block, choose $T_2$ in the second layer, and so on, forming $g^l$ different models. As we explicitly optimized only for $g$ models, instead of the exponentially many models, are the extracted models accurate? Surprisingly, in multiple settings, and for a various model sizes, we observe that the extracted models indeed are accurate, with accuracy scaling with the size of the extracted model.

We train Matformer-based decoder-only Language Models (MatLM) up to 2.6B parameters and observe that: (a) MatLMs explicitly trained with $g$ exponentially spaced granularities almost match validation loss and one-shot downstream evals of respective $g$ baseline models trained independently from scratch, (b) our extracted models using Mix'n'Match lie on the accuracy-vs-parameters trade-off curve generated by the $g$ explicitly trained models, (c) through scaling experiments we observe that the loss vs compute law for different MatFormer models remains similar to vanilla Transformer models across different granularities and (d) the submodels extracted from MatLM have highly consistent behavior that is highly desirable for inference optimizations and deployment across scales.

We further studied MatFormer-based ViT models (MatViT) and have similar observations as MatLM. For example, MatViT-L/16 improves the accuracy of the standard ViT-L/16 model on ImageNet-1K, and the extracted sub-models all match or even perform better than the independently trained baselines. Furthermore, we demonstrate that, due to high consistency, MatViT models can be used as "elastic encoders" for adaptive image retrieval. That is, the metric-space of an image encoded by the universal (i.e. the largest) MatViT model is roughly preserved by the nested submodels. Hence, based on query complexity, system load, and various other considerations, we can use one of the extracted MatViT encoders at inference time for retrieval on a fixed corpus encoded by the universal model – providing over $40\%$ lesser compute overhead with $< 0.5\%$ drop in accuracy.

**We make these key contributions:**

1. We introduce MatFormer, which incorporates a nested sub-structure within the standard Transformer and jointly optimizes all the $g$ granularities to produce a single, universal elastic model.

2. Employing Mix'n'Match of granularities across layers in a universal MatFormer model yields hundreds of accurate and consistent submodels without any additional training cost (Section 3).

3. MatFormer generalizes effectively to both decoder-only language models (MatLM) and vision encoders (MatViT), scaling as reliably and accurately as the standard Transformer, while enabling significantly faster autoregressive generation and large-scale adaptive dense retrieval (Section 4).

## 2    Related Work

A standard Transformer (Vaswani et al., 2023) has become the unifying model architecture for foundation models (Bommasani et al., 2021) across modalities like language (Brown et al., 2020), vision (Dehghani et al., 2023) and audio (Radford et al., 2023). While extremely powerful, the standard Transformer block is not natively elastic in a way that enables large-scale adaptive and flexible deployment across various resource constraints. To cater to the plethora of deployment requirements, existing solutions include training a family of models of varying sizes (Anil et al., 2023; Touvron et al., 2023b), post-hoc efficiency techniques like quantization (Dettmers & Zettlemoyer, 2023), pruning (Lagunas et al., 2021), distillation (Sanh et al., 2019) and mixture of varying capacity experts (MoE) (Zhang & Ma, 2012). However, these solutions often are specific to the single constraint at hand, and require additional training or trade-off memory/compute during inference making them far from being a truly elastic solution for adaptive deployment. Lastly, Transformer based LLMs are often sped-up during inference with techniques like speculative decoding (Leviathan et al., 2023; Chen et al., 2023) – that benefits from the smaller draft & the larger verifier models having similar behavior – or early exiting (Schuster et al., 2022) to enable real-time deployment.

Obtaining multiple smaller models from a single model has been explored in the past (Yu et al., 2018; Yu & Huang, 2019; Cai et al., 2019; Grimaldi et al., 2022; Cai et al., 2021) with most works focusing on CNN encoders. Specifically, OFA (Cai et al., 2019) creates a universal CNN model which is used to extract and finetune submodels for a handful of deployment constraints while slimmable networks (Yu et al., 2018) optimize for limited preset widths and require explicit training to interpolate for a few more intermediate widths (Yu & Huang, 2019). NAS techniques that sample random (not nested) subnetworks during training at each step, and then find the subnetwork architecture to retrain from scratch before deployment have been explored (Wang et al., 2020b). These techniques fall short of being truly elastic and come with significant training overheads. More recently some of them have been extended to Transformer encoders (Chavan et al., 2022; Hou et al., 2020; Salehi et al., 2023) for extracting sub-models in both static or dynamic settings but fail at extending further to decoder-only language models. While not in the weight space, matryoshka representation learning (Kusupati et al., 2022) & FlexiViT (Beyer et al., 2023) showcase elasticity in output & input spaces respectively by smoothly spanning deployment constraints with minimal overhead. MatFormer, in contrast, builds upon these works by nested the weight space instead to enable truly

elastic and adaptive Transformer-based (decoder & encoder) models that span all the accuracy-vs-compute tradeoff (statically or dynamically) with minimal changes and training overhead (Figure 1). Finally, we also point the readers to SortedNet (Valipour et al., 2023), a concurrent work with similar goals applied to encoders, which optimizes many sampled submodels (akin to prior works) unlike MatFormer's joint optimization of a few (typically 4) nested submodels.

## 3   MatFormer

In this section, we define MatFormer's nested substructure (Section 3.1) and discuss its training procedure for a chosen $g$ model granularities (Section 3.2). We then discuss elastic inference using Mix'n'Match models (Section 3.3) from MatFormer along with its deployment considerations.

### 3.1   MatFormer Structure

MatFormer defines $g$ Transformer blocks $T_i$, such that, $T_1 \subset T_2 \subset \cdots \subset T_g$ where $T_i \subset T_{i+1}$ indicates that the parameters of $T_i$ are contained in those of $T_{i+1}$. While it is possible to impose such a structure on any part of the Transformer, we select the FFN block to define our method and present our experiments, as the model size and computational cost of a Transformer is dominated (around $60\%$ for LLMs and ViTs) by the FFN block (see Appendix B).

The Transformer FFN block has a single hidden layer with $d_{\text{ff}}$ neurons and both input and outputs in $\mathbb{R}^{d_{\text{model}}}$, and fixed FFN ratio := $d_{\text{ff}}/d_{\text{model}}$ (typically $\geq 4$). MatFormer introduces the matryoshka nested structure with $g$ granularities on the hidden representation of the FFN block. Concretely, a nested sub-block of the Transformer, $T_i$ contains the first $m_i$ neurons of the FFN and $1 \leq m_1 \leq \cdots \leq m_g = d_{\text{ff}}$ represent the number of neurons for each granularity or sub-model. So, depending on the chosen granularity the FFN operation of $T_i$ i.e., $T_i^{\text{FFN}}$ on an input $x \in \mathbb{R}^{d_{\text{model}}}$ is:

$$T_i^{\text{FFN}}(x) = \sigma(x \cdot \mathbf{W}_1[0:m_i]^\top) \cdot \mathbf{W}_2[0:m_i], \qquad (1)$$

where the weight matrices of FFN are $\mathbf{W}_1, \mathbf{W}_2 \in \mathbb{R}^{d_{\text{ff}} \times d_{\text{model}}}$ and bias terms are omitted for simplicity. $\mathbf{W}_1[0:k]$ denotes the submatrix with the first $k$ rows of $\mathbf{W}_1$. Finally, $\sigma$ is a non-linearity often set to GELU (Hendrycks & Gimpel, 2016) or squared ReLU (So et al., 2021). In this work, we chose the $g = 4$ exponentially spaced granularities with FFN ratios of $\{0.5, 1, 2, 4\}$ i.e., the nested hidden neurons are of the sizes $\{\frac{d_{ff}}{8}, \frac{d_{ff}}{4}, \frac{d_{ff}}{2}, d_{ff}\}$.

With the nested MatFormer blocks $T_1, T_2 \ldots T_g$, we can combine these to form a MatFormer model, with $g$ nested submodels $\mathcal{M}_1 \subset \mathcal{M}_2 \ldots, \subset \mathcal{M}_g$ where $\mathcal{M}_i \leftarrow [T_i]^{\times l}$, i.e., $\mathcal{M}_i$ is formed by stacking $T_i$ for $l$ layers. The input and output embedding matrices are shared across the models.

### 3.2   Training

For a Transformer model $\mathcal{M}$, the forward pass on an input $x$ is denoted by $\mathcal{M}(x)$ and let $\mathcal{L}$ denote the loss function between the output and the target $y$: $\mathcal{L}(\mathcal{M}(x), y)$.

MatFormer relies on a simple training strategy of jointly optimizing all the $g$ nested submodels together. To this end, we set the MatFormer loss as a weighted average of loss of $g$ submodels and train for it using the standard stochastic gradient-based optimizers (Shazeer & Stern, 2018):

$$\mathcal{L}_{\text{JOINT}}(x, y) = \sum_{i=1}^{g} \lambda_i \cdot \mathcal{L}(\mathcal{M}_i(x), y), \qquad (2)$$

where $\lambda_i > 0$ is the weight of $i$-th granular submodel. In this paper, we set $\{\lambda_i\}_{i=1 \ldots g}$ to be uniform i.e., $1/g$ but explore tuning $\{\lambda_i\}_{i=1 \ldots g}$ in Appendix E.4 to further improve MatFormer.

The joint training in MatFormer involves one forward pass per each of the $g$ submodels and benefits from portions of shared computation during backpropagation. MatFormer training results in $g$ accurate nested submodels $\mathcal{M}_{1 \ldots g}$ inside the universal MatFormer model ($\mathcal{M}_g$). Note that this simple strategy outperforms various other training techniques (Appendix E.2). Finally, instead of pretraining models with MatFomer structure, we can also induce this structure via finetuning.

MatFormer training is $\sim 15\%$ faster (for $g = 4$) than training all the Transformer based equivalent submodels independently (Appendix B). However, MatFormer also enables the extraction of

hundreds of smaller submodels along the accuracy-vs-compute curve traced by the $g$ explicitly optimized submodels (Section 3.3). These models emerge for free using Mix'n'Match during inference and drastically reduce the amortized training cost per model obtained through MatFormer. The joint optimization, even without self-distillation from $\mathcal{M}_g$, results in smaller submodels that have highly consistent behavior (Section 3.4) with the universal model. Finally, in Appendix B.1, we argue that the training efficiency of MatFormer can be significantly improved through various optimizations.

### 3.3 Mix'n'Match

At inference time, it is trivial to extract one of the $g$ submodels $\mathcal{M}_1 \subset \mathcal{M}_2 \ldots, \subset \mathcal{M}_g$ by stacking the corresponding Transformer block $T_i$ across layers. However, by selecting different granularity for each MatFormer layer, it is possible to generate a combinatorially large number of accurate smaller models for free. We call this simple procedure *Mix'n'Match* and observe that these additional model granularities –which were never explicitly optimized – are highly performant.

In fact, we can further increase the number of extracted models by generating interpolating blocks between fixed granulaties (Kusupati et al., 2022). For example, we can generate a $\widetilde{T}$ block that uses first $\frac{1}{2}(m_i + m_{i+1})$ neurons in the FFN layer which still tends to be highly accurate.

To summarize, given a computational budget, we can extract a highly accurate model with Mix'n'Match for the constraints rather than using a smaller less accurate model or training a model for this specific constraint (Sections 4.1.1 & 4.2). We note that a compute constraint can be satisfied by various Mix'n'Match models with different accuracies, making identifying the best Mix'n'Match configurations without downstream validation is an exciting direction for future work.

### 3.4 Deployment

During deployment, all we need to store is the single universal MatFormer model for different types of elastic inference depending on the constraints. In the case of static workloads, where compute resources are known beforehand and the inputs remain relatively similar in difficulty, one can choose the most accurate static submodel for the constraints using Mix'n'Match. This eliminates the usage of a less accurate preexisting model or training of a new one for the specific constraints.

For dynamic workloads, where the compute resources or the input hardness change on the fly, we can use the universal MatFormer model to dynamically extract the optimal submodel for token-based routing in LLMs akin to MoE (Kudugunta et al., 2021; Li et al., 2022) and elastic encoders in dense retrieval (Section 4.2.2). This works largely because all the extracted submodels have high behavioral *consistency* with universal MatFormer model (Section 4.1) – minimizing the drift across predictions from various submodels. We measure the consistency between two generative models as the *percentage of matching tokens* generated by them for the same prefix or using the *KL divergence* of the smaller model outputs with the larger model outputs – this accounts for potential sampling strategies in decoding. This highly consistent nature of MatFormer results in superior inference time speedups for techniques like speculative decoding (Leviathan et al., 2023) (Section 4.1.1) and can assist in reducing prediction drift between cross platform deployments. We also show that higher model consistency also aids metric-space structure preservation in encoder models (Section 4.2.2).

## 4 Experiments

In this section, we empirically evaluate MatFormer across modalities (language and vision), model classes (decoder and encoder), and scales (up to 2.6B parameters). Specifically, we train and analyze MatFormer-based decoder-only Language Models – MatLMs (Section 4.1) – and encoder-only Vision Transformers – MatViT (Section 4.2) models with $g = 4$ nested granularities across various model sizes. For a fair comparison, we also independently train the Transformer baseline for the submodel of each granularity across model sizes for the same tasks. We primarily focus on the elastic deployment of MatFormer-based models (Sections 4.1.1 & 4.2) for tasks spanning from one-shot generative evals to adaptive image retrieval. Additionally, we also investigate the reliable scaling behavior (Kaplan et al., 2020) of the MatFormer models (Section 4.1.2).

### 4.1 MatLM: MatFormer Language Models

We build MatFormer-based decoder-only Language Models – MatLMs – and contrast them to their vanilla Transformer counterparts (LMs) (Liu et al., 2018). The LMs broadly follow the training

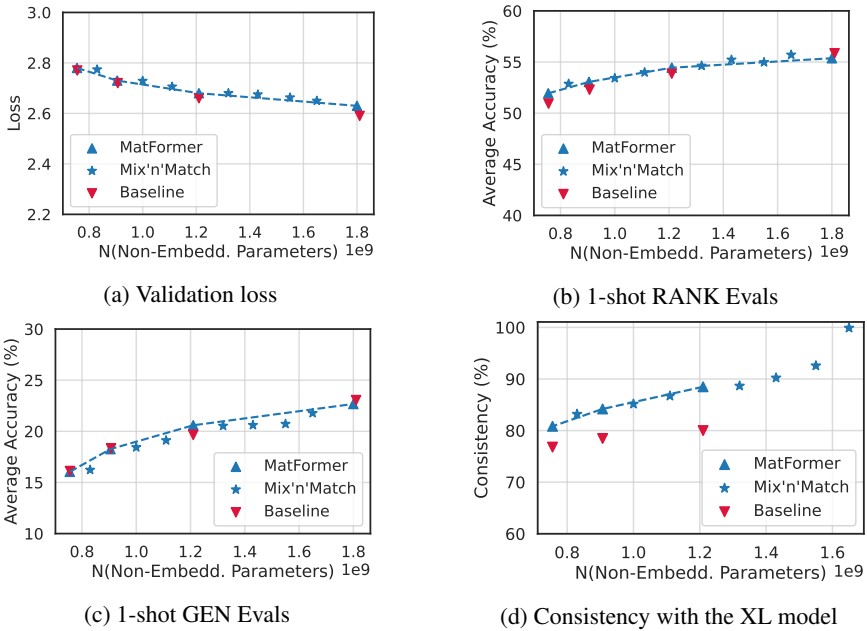

|   |   |
|---|---|
| (a) Validation loss | (b) 1-shot RANK Evals |
| (c) 1-shot GEN Evals | (d) Consistency with the XL model |

Figure 2: Validation loss & one-shot downstream evaluation scores for the 2.6B MatLM & baseline models. Mix'n'Match helps generate accurate and more consistent models from MatLM that lie on the performance-vs-compute curve spanned by the explicitly optimized submodels.

pipeline and procedure outlined by Thoppilan et al. (2022). For each MatLM model with a set $d_{model}$, we jointly optimize for $g = 4$ nested granularities represented by FFN ratios of $\{0.5, 1, 2, 4\}$ – i.e., only the hidden representation size of the FFN block changes. We denote these submodels as MatLM – {S, M, L, XL} in increasing order of model size and refer to MatLM-XL as the universal MatLM. For baselines, we train vanilla Transformer models with comparable architectures. That is, for each MatLM, we train 4 separate baseline models with FFN ratios of $\{0.5, 1, 2, 4\}$ for a fixed $d_{model}$ denoted as Baseline – {S, M, L, XL}. We evaluate these models on validation loss (= log perplexity) and average accuracy on 26 English tasks similar to (Brown et al., 2020; Du et al., 2022; Anil et al., 2023). Of these 26 tasks, we group 5 tasks that require generating multiple tokens under "GEN" and the remaining tasks that involve choosing an option from the input text under "RANK". Please see Appendix A for further details on training, evaluation, and the datasets.

#### 4.1.1 Elastic Inference with MatLM

To showcase elastic inference, we evaluate the 2.6B parameter MatLM models on its ability (a) to provide models spanning the accuracy-vs-compute curve using Mix'n'Match (Section 3.3) and (b) to improve post-hoc inference optimization techniques like Speculative Decoding (Leviathan et al., 2023) to further speed-up accurate auto-regressive generation.

**Accurate MatLM submodels for every constraint for free with Mix'n'Match.** Leveraging Mix'n'Match, a MatLM can provide accurate models for every compute constraint (between S and XL), not just the explicitly optimized granularities {S, M, L, XL}. We evaluate the impact of Mix'n'Match on the 2.6B parameter MatLM in Figure 2 through validation loss and downstream evals and contrast them to four granularities {S, M, L, XL} of the 2.6B baseline LM (all trained independently). In Figures 2a, 2b & 2c, we show that all MatLM – {S, M, L, XL} models all perform as well as their corresponding baselines – with marginal improvements and drops across the scale.

In Figure 2a we see that Mix'n'Match helps obtain many models on the optimal loss-vs-compute curve at zero cost. Moreover, downstream eval tasks on these Mix'n'Match models also mimic this trend, as shown in Figures 2c & 2b. In a deployment setting that only has 55% of the required compute resources needed for the MatLM-XL model, it is now possible to have a Mix'n'Match submodel with < 2% accuracy drop on RANK evals. Without elastic deployment due to Mix'n'Match, we would see a > 2.5% accuracy drop due to the use of the MatLM-M model. Note that we highlight only a few of the hundreds of accurate Mix'n'Match models along the curves. We discuss additional details and results on the Mix'n'Match procedure in Appendix C.

**MatLM submodels speed up speculative decoding.** Speculative decoding leverages an accurate lightweight LM as a draft model to autoregressively generate a few tokens, followed by verifying these drafts with a larger model through parallel decoding on the generated tokens. When the draft is inaccurate, the draft model is rolled back and reset to the larger model's output. This results in considerable inference speed-up for the *same accuracy as the large model*. We point the reader to the original paper for a more detailed explanation (Leviathan et al., 2023).

Slow down of this algorithm stems from cases where the smaller model's predictions disagree with the larger model. A draft model that is significantly more consistent with the larger verifier model would lead to less rollbacks of the draft predictions and therefore lower latency. As seen in Figure 2d the MatLM submodels can be up to 8.5% more consistent than the baselines to their corresponding XL model. The significant gap persists even in the KL divergence variant of consistency with the XL model's outputs (see Figure 8 in Appendix). This improved consistency along with the need for only a single universal model positions MatLM favorably to improve techniques that require draft and verifier models such as speculative decoding.

Table 1: Inference time speed-ups over a standard 2.6B model through speculative decoding using a 1.5B (S) draft and 2.6B (XL) verifier model.

| Speculative Decoding | LAMBADA | TriviaQA |
|---|---|---|
| Baseline | $1.10\times$ | $1.08\times$ |
| MatLM | $1.14\times$ | $1.11\times$ |
| + shared attention cache | $1.16\times$ | $1.14\times$ |

Table 1 shows the inference time speed-ups from speculative decoding using the S and XL submodels of the 2.6B language model for drafting and verification respectively. Speculative decoding with independently trained baseline LMs results in a speed-up of up to 10% over the standard autoregressive decoding of the 2.6B-XL model. But MatLM-based speculative decoding is up to 6% faster than traditional speculative decoding. This additional speed-up can be primarily attributed to the more consistent nature of MatLM-based drafter and verifier models and is further boosted by the ability to share attention cache across models from MatLM which is infeasible for the baselines (see Appendix B.2). Finally, MatLM further reduces the memory overhead for inference by removing the need to have two models during resource-constrained deployment.

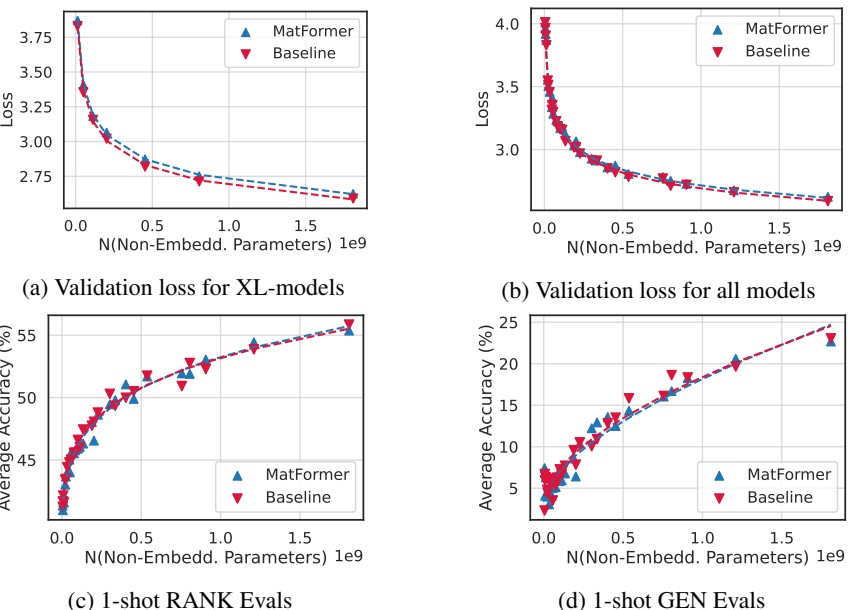

(a) Validation loss for XL-models

(b) Validation loss for all models

(c) 1-shot RANK Evals

(d) 1-shot GEN Evals

Figure 3: We train decoder-only MatLM models at a range of sizes from 78M to 2.6B parameters and observe the scaling trends of all granularities (S, M, L, XL) for validation loss and 1-shot downstream evaluation scores. We find that the MatLM-XL models across scales mimic the training trends of Baseline-XL models. Interestingly, we also note that that validation loss and downstream evaluations follow the *scaling trends of the XL-models across all granularities*.

#### 4.1.2 MatLM Scales as well as Vanilla Transformer LMs

Now that we have established that a 2.6B MatLM model and its submodels are as accurate as the baseline Transformer LMs, we want to examine the scalability of training MatLM models. So, we study the scaling properties (Kaplan et al., 2020; Hoffmann et al., 2022) of MatLMs and compare them to vanilla Transformer baseline LMs trained for the same number of tokens. We train models ranging from 78M to 2.6B parameters on 10B to 160B tokens and plot the validation loss for MatLM – {S, M, L, XL} compared against their baselines in Figure 9.

First, in Figure 3a, we observe that the training of MatLM-XL models across model sizes scale as reliably as the Baseline-XL LMs for loss vs. number of parameters. However, Figure 3b interestingly shows that it is not just the XL models but rather all the nested submodels, irrespective of granularity {S, M, L, XL}, of MatLM and Baseline that follow the same scaling trend. Therefore, we fit a scaling law according to the number of non-embedding parameters ($N$) and training tokens ($D$) for all possible submodels for both MatLMs and the baselines in Table 2. We observe that the fitted parameters are extremely similar, suggesting that MatLMs scale similarly to vanilla Transformer LMs.

In Figures 3c & 3d we also find that the downstream evals for MatLM are within $0.5\%$ of the baselines, with the smaller submodels even outperforming the baselines at scale. Finally, Figure 9f in the Appendix shows that the MatLM submodels are more consistent with their XL model compared to the baseline counterparts across scales.

Table 2: Fitted parameters for the scaling equation: $\text{Loss}(N, D) = a \cdot (ND)^b + c$

|  | a | b | c |
|---|---|---|---|
| Baseline | 20.917 | -0.119 | 1.868 |
| Matformer | 17.516 | -0.114 | 1.845 |

We note that the scaling law equation does not capture how (1) MatLMs have been optimized for multiple submodels and even have performant submodels that have not been explicitly optimized for (Section 4.1.1), and (2) MatLMs and baselines of the same size have different training FLOPs per step. We leave formulations that capture these subtleties to future work and further discuss this in Appendix D.1. We provide full results split by granularity in Appendix D.

### 4.2 MatViT: MatFormer Vision Transformers

In this section, we extend MatFormer to Vision Transformer (ViT) (Dosovitskiy et al., 2020) based computer vision encoder models. MatFormer-based ViT – MatViT – enables elastic inference for fundamental tasks like image classification and retrieval. To this end, we train the MatFormer variant of the standard ViT-B/16 and ViT-L/16 models – MatViT-B/16 and MatViT-L/16 that are trained with $g = 4$ prechosen nested granularities (FFN ratios of $\{0.5, 1, 2, 4\}$). B/16 models are trained on ImageNet-1K (Russakovsky et al., 2015) with AugReg (Steiner et al., 2021) while L/16 models are pretrained on ImageNet-21K (Deng et al., 2009) followed by finetuning on ImageNet-1K. All models are trained with the training setup and optimal hyperparameters of the standard ViT variants from the Scenic library (Dehghani et al., 2022).

#### 4.2.1 Image Classification

For image classification, we evaluate both ViT & MatViT models on ImageNet-1K. Figure 4a shows that the explicitly optimized granularities in MatViT result in as accurate models as the independently trained baselines for the B/16. However for L/16, as shown in Figure 4b, we see that the MatViT models are up to $0.35\%$ more accurate than the baseline for the same inference cost.

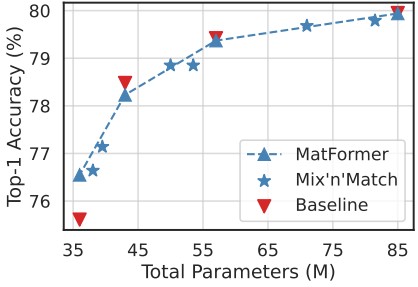

(a) B/16 trained on ImageNet-1K with AugReg

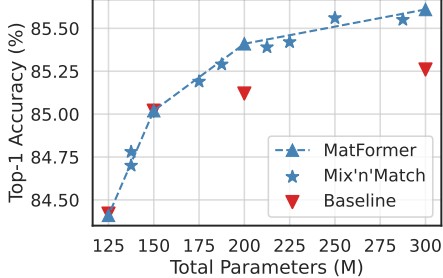

(b) L/16 pretrained on IN-21K → ImageNet-1K.

Figure 4: MatViT variants match or outperform standard ViT models on ImageNet-1K classification and provide free extracted models that span the accuracy-compute curve through Mix'n'Match.

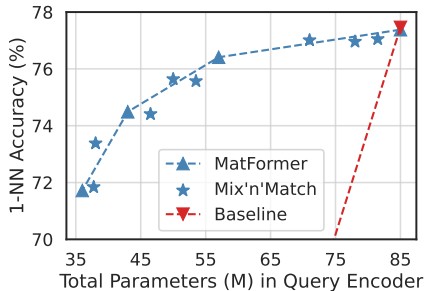
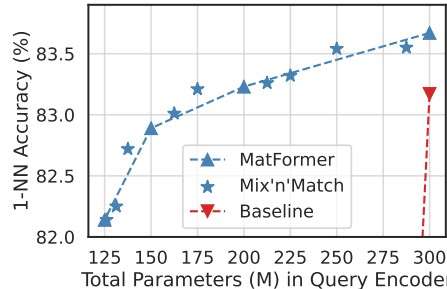

(a) B/16 trained on ImageNet-1K with AugReg

(b) L/16 pretrained on IN-21K → ImageNet-1K.

Figure 5: MatViT natively enables elastic encoders for adaptive retrieval that can be used for real-time query side computation while retaining strong accuracy on ImageNet-1K, unlike the baselines.

We then explore using MatFormer at different training stages with a $2 \times 2$ grid of pretraining-finetuning pairs (Table 7 in Appendix F.1) and find that using a MatFormer during pretraining helps bring more accurate and flexible encoders for downstream use. Further, finetuning using MatFormer enhances elastic deployment depending on the constraints at hand through Mix'n'Match.

**Adaptive Encoders with Mix'n'Match.** Furthermore, our Mix'n'match models' accuracy almost lies on the line joining accuracy of explicitly trained granularities. In scenarios where, say, an application can host 50M parameter B/16 model, MatViT can provide $0.8\%$ more accurate model than the current approach which would host the largest baseline model with $\leq$ 50M parameters.

During deployment, the universal MatViT model can be stored in memory and depending on the compute constraints be used to extract an adaptable smaller model to maximize accuracy with the available resources at that moment. Currently, we find the Mix'n'Match models on the accuracy-compute curve through a quick inference on the validation set. While relatively scalable, this points to the need for optimal budget allocation across layers in neural networks (Kusupati et al., 2020).

### 4.2.2 Adaptive Image Retrieval

The goal of image retrieval is to find semantically similar images – e.g. images from the same class – using representations obtained from a pretrained encoder (Chen et al., 2022). Standard approach is to encode the database images as well as query image with same encoder and run nearest neighbor retrieval for the query embedding. While we can embed database images with an expensive encoder, the query encoder generally has to be real-time. Furthermore, the setting of query encoding might be varied, e.g., on-device vs. cloud processing, varying query load and query complexity. Current solutions have to stick to a fixed encoder thus compromising on accuracy or cost for various settings.

Given the elastic nature of MatViT, it is a good candidate for query encoder. However, retrieval also requires that submodels preserve distances between fixed database (with large encoder) and query embeddings across all the granularities. If we use smaller baseline ViT models only for query encoding, these distances are not preserved and lead to nearly 0 retrieval accuracy (see Figure 5).

We evaluate both ViT and MatViT encoders on ImageNet-1K for image retrieval. We compute 1-nearest neighbor (NN) accuracy using the representation vector of the [CLS] token (also see Appendix F.2). Figure 5 shows that submodels extracted from MatViT can approximately preserve distances and provide significantly more flexibility. For example, with a loss of $< 0.5\%$ accuracy, MatViT-L/16 can reduce compute cost by $40\%$. To our knowledge, this is the first result of its kind and opens up a wide variety of adaptive inference strategies for large-scale semantic search.

## 5 Conclusions

In this work we presented MatFormer, a natively elastic Transformer architecture that allows training a single universal model which can be used to extract hundreds of smaller accurate submodels at zero additional cost at deployment time We find that the MatFormer Language Model (MatLM) matches the perplexity & 1-shot accuracy of independently trained models. In fact, MatLM demonstrates an interesting loss-vs-compute scaling curve that is nearly *independent* of trained granularity indicating robust generalization to *extremely* large models as well. Finally, MatFormer submodels enable diverse inference time speedups like faster autoregressive generation with speculative decoding and elastic query encoders for adaptive dense retrieval across modalities.

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

# A  Implementation Details

## A.1  Architecture and Training

For our experiments, we train a range of MatLMs varying from size 78M to 2.6B for 10B-160B tokens – we scale model size equally with the number of training tokens (Hoffmann et al., 2022). For each MatLM granularity, we also train a corresponding baseline vanilla Transformer model. That is, for each model size we train Baseline-XL, L, M, S with $d_{ff} = 4 * d_{model}, 2 * d_{model}, d_{model}, d_{model}/2$. All models have 16 layers, 16 attention heads, and a $d_{model} : d_{ff}$ ratio of $1 : 4$. We train a 256k vocabulary using the SentencePiece library (Kudo & Richardson, 2018), use a maximum context length of 1024 tokens, and a batch size of 1M tokens. We pretrained the 2.6B on 256 v3 TPU chips. We provide further details on these models in Table 3. For further details on training data, we point the reader to (Thoppilan et al., 2022).

Table 3: Model details for the models scales used to conduct the experiments described in Section 4.1, with a breakdown of total parameter counts, non-embedding parameter counts and FFN parameter counts for each model granularity.

| Parameter Count (full / spliced) | Non-Embedding Params (full / spliced) | FFN Params (full) | $d_{\text{model}}$ | N(tokens) |
|---|---|---|---|---|
| 78M (74M / 72M / 71M) | 12.6M (8.4M/6.3M/ 5.3M) | 8.4M | 256 | 10B |
| 180M (164M / 157M / 152M) | 50M (33.7M/25.3M/21.1M) | 33.6M | 512 | 20B |
| 310M (272M / 253M / 244M) | 113M (75M/56M/47M) | 75.6M | 768 | 30B |
| 463M (397M / 363M / 346M) | 201M (134M/100M/84M) | 134M | 1024 | 40B |
| 850M (696M / 620M / 582M) | 453M (302M/227M/189M) | 302M | 1536 | 80B |
| 1.3B (1B / 927M / 860M) | 805M (537M/403M/335M) | 537M | 2048 | 120B |
| 2.6B (2B / 1.7B / 1.54B) | 1.8B (1.2B/0.9B/0.7B) | 1.2B | 3072 | 160B |

## A.2  Downstream Evaluation

We evaluate all the LM models trained on set of 26 English tasks similar to (Brown et al., 2020; Du et al., 2022; Chowdhery et al., 2022; Anil et al., 2023), including:

1. **Open-Domain Closed-Book Question Answering tasks**: TriviaQA (Joshi et al., 2017), Natural Questions (Kwiatkowski et al., 2019), and WebQuestions (Berant et al., 2013).
2. **Cloze and completion tasks:** LAMBADA (Paperno et al., 2016), HellaSwag (Zellers et al., 2019), and StoryCloze (Mostafazadeh et al., 2016).
3. **Winograd-style tasks:** Winograd (Levesque et al., 2012) and WinoGrande (Sakaguchi et al., 2019).
4. **Reading comprehension:** SQuAD v2 (Rajpurkar et al., 2018) and RACE (Lai et al., 2017).
5. **Common sense reasoning:** PIQA (Bisk et al., 2019), ARC (Clark et al., 2018), and Open-BookQA (Mihaylov et al., 2018).
6. **SuperGLUE** (Wang et al., 2020a)
7. **Natural language inference:** Adversarial NLI (Nie et al., 2020).

Among all the downstream datasets, we classify LAMBADA, Natural Questions, SQuAD v2, WebQuestions, and TriviaQA under "GEN" tasks as these require generating a few tokens, and the remaining tasks under "RANK" tasks as they consist of choosing an option among the choices given along with the input. For all the granularities corresponding to each model, we present evaluation numbers along with development set log perplexity loss on all the 26 tasks in Tables 9 to 15. We also perform evaluation on 2.6B Mix'n'Match models and provide it in Table 16.

# B  Training and Inference Costs

We currently make minimal changes and optimizations to the training scripts of vanilla Transformer architecture. In other words, we use the same code for both Baselime and MatFormer, except using different sized splices of FFN block for each forward pass. Note that this implementation is suboptimal, as it involves added communication costs of FFN weight matrices when using model parallel

Table 4: 2.6B MatLM and Baseline training time per step, GFLOPs per step, and forward pass latencies. Each model is trained on 256 v3 TPU chips. Note that MatLM Fwd pass latency for any granularity will be same as corresponding Baseline granularity latency.

| Model | Time (s) / step | GFLOPs / step | Fwd pass latency (s) |
|---|---|---|---|
| MatLM | 2.326 | 470841 | - |
| Baseline-XL | 0.728 | 186884 | 0.234 |
| Baseline-L | 0.670 | 147317 | 0.215 |
| Baseline-M | 0.652 | 125517 | 0.198 |
| Baseline-S | 0.630 | 117556 | 0.190 |

training (discussed in more details in Appendix B.1). Though using a suboptimal implementation, we achieve the wall-clock time for MatLM training $\sim 15\%$ less to sum of wall-clock times to train all the $4$ granulatities baseline counterparts. We also note that at train time, the peak memory usage is roughly equal to the sum of memory usage for the independently trained baselines. On the other hand, at inference time, both baseline and MatFormer have the same memory footprint. We give exact FLOP count, wall-clock time, and forward pass time (inference cost) of each baseline and MatLM 2.6B model (or its corresponding smaller granularities) in Table 4. During serving, we observe the 2.6B model FFN latency to attention latency ratio $= 56 : 44$. We note that this FFN:MHA latency ratio depends highly on scale and sequence length. More specifically, for a given sequence length FFN latency dominates the overall latency at scale, while the attention heads' cost increases with sequence length. We refer the reader to Kim et al. (2023) for a more extensive illustration of this. We emphasize that though we trained one MatFormer and compare its training time with Baselines combined, we get many more models than the 4 model granularities we explicitly trained.

## B.1 Improving MatFormer Training Efficiency

While MatFormer training uses asymptotically $2\times$ FLOPs compared to a regular Transformer, optimizations are necessary to also realize a $2\times$ runtime training performance. We discuss a few strategies here, leaving exact experimental testing to future work.

**Delayed gradient synchronization via local accumulation.** Since multiple forward and backward passes are made for each mini-batch in common implementations of data parallelism, this induces a gradient synchronization across all device for each backward pass with additional gradient accumulation. As such, for MatFormers a minimum of $2\times$ the parameters worth of gradients are exchanged for the MLP layers, thus increasing the communication overhead. Additionally, for some frameworks, such as PyTorch, gradients of the full-weight matrix size need to be exchanged, leading to $4\times$ more communication for our default experimental setup. A more efficient way to communicate gradients is to keep a local gradient accumulation buffer, which is used to accumulate all gradient from all subnetworks into the main, full-sized weight gradient. After all forward-backward passes have been completed, synchronization of gradients – with additional overall of computation and communication – can ensue. This saves $2\times$ communication overhead, reducing communication overhead to the same cost as a regular Transformer.

**Fused MatFormer kernels.** Depending on the accelerator (GPU/TPU), the smallest MatFormer forward and backward pass can be inefficient in that the matrices are too small to fully utilize the accelerator. To improve utilization at the cost of additional memory for activations, it is possible to run the following computational fusion strategy for MatFormer computation: (a) duplicate mini-batch $4\times$, (b) do the forward/backward pass for each layer for all MatFormer stages at the same time, (c) in doing so, load the tile for the weight matrix once, and reuse it for all relevant MatFormer stages. This strategy is similar to tiling strategies in FlashAttention (Dao et al., 2022) or convolution (Krizhevsky, 2009) which increase the arithmetic intensity for small weights by reusing of matrix multiplication tiles written to SRAM.

## B.2 Speculative Decoding Attention Sharing

An additional benefit of MatLM is that the attention cache is shared between the draft and verifier model. When the XL model verifies S model's draft, it overwrites the attention cache with its richer latent representation compared to the one generated by the drafter model. Note that 1) this does

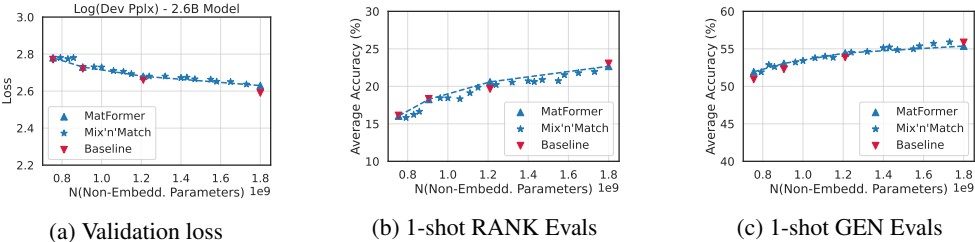

| (a) Validation loss | (b) 1-shot RANK Evals | (c) 1-shot GEN Evals |

Figure 6: Validation loss & one-shot downstream evaluation scores for the 2.6B MatLM & baseline models. Mix'n'Match helps generate accurate models from MatLM that lie on the performance-vs-compute curve spanned by the explicitly optimized submodels.

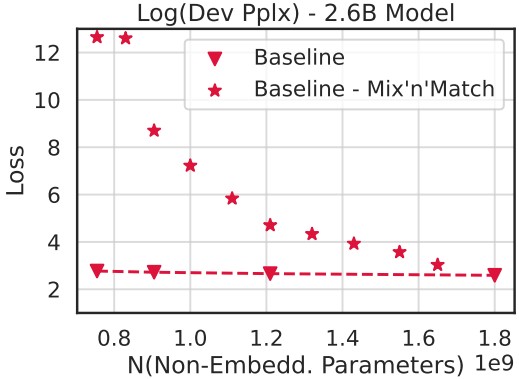

Figure 7: Validation loss for the 2.6B baseline models and their Mix'n'Match counterparts. Unlike MatLM, these extracted subnetworks perform poorly.

not involve extra computation since MatLM has a single universal model including both draft and verifier model; 2) attention sharing isn't possible in the Baseline since they are not explicitly trained together. Hence, latent representation of one model is quite meaningless to the other model. Thus, attention sharing gives further improvement over vanilla speculative decoding as shown in Table 1.

## C   Mix'n'Match

To implement Mix'n'Match, we experimented with several heuristics to select the best subnetwork, but consistently observed that gradually using larger granularities in deeper layers worked the best. More formally, we use non-decreasing hidden dimensions with the least slope (change in hidden dimensions across consecutive layers) across layers. Given that this choice behaves nearly optimally (performance lies on the pareto-optimal curve), we did not focus on search techniques. For completeness, we have plotted additional extracted subnetworks (in addition to what we have plotted in Figure 2) in Figure 6. These additional datapoints follow a similar trend. In Figure 7, we plot the validation loss of applying Mix'n'Match to vanilla Transformer baselines, and find the ability to Mix'n'Match granularities is restricted to MatLMs. In future work, we plan to extend the nested substructure to other components of the Transformer - attention heads, model dimensions, and n(layers). This would combinatorially expand the search space, warranting the use of more advanced search methods. We leave this exploration to future work.

## D   Scaling Laws for Language Decoders

We provide results split by granularities for validation loss, average score on RANK tasks, average score on GEN tasks, and consistency in Figures 9, 10, 11, and 12 respectively. We observe that while the gap in validation loss between MatLMs and Baselines appears to be constant, the gap for downstream evaluations reduces with scale - in fact, granularities L, M and S have better downstream performance for models larger than 1B. For consistency, the gap appears to reduce with scale, but one would need to scale the models by many orders of magnitude beyond what's possible today for baselines to have comparable consistency with MatLMs.

### D.1 Scaling laws of MatFormers vs Transformers.

Scaling laws are essential tools to estimate optimality under as the cost of training or inference is increased. Scaling laws can take diverse viewpoints such as overall training cost in FLOPS, training data and parameter efficiency, and inference mean FLOPS utilization vs latency for deployments.

The scaling relationship of MatFormers versus Transformers is both simple and complex. Simple, because MatFormers scaling curves for pretraining are only slightly offset from Transformers – thus MatFormers only require a fixed relative amount of additional compute and the same hyperparameters that work for Transformers are effective for MatFormers. For the setting where we use the same hyperparameters as Transformers, MatFormers need at most $10 - 20\%$ more training tokens to reach the same loss as a regular Transformer. Initial experiments where we tune hyperparameters for the individual forward/backward passes and by performing more careful initialization of the subslices the gap appears to shrink. While we do not have enough data to make definite statements, it appears MatFormer scaling can be improved to be close to Transformers scaling needing less than $0 - 5\%$ additional training tokens.

The complex scaling relationship comes from the fact that MatFormers allow the training of multiple models with a single training run which is a qualitative different from Transformers and difficult to factor into scaling equations. Essentially, in terms of efficiency, if we compare the training FLOPs equivalent of all the extractable models from MatFormers, then MatFormer training alone has a clear advantage in any case where all parameters used to train standard Transformer models on the same dataset exceed $2.58P$, where $P$ is the number of parameters of the MatFormer and the largest Transformer model. This is so because MatFormers use 2.58 times more FLOPs per token for a training run than a Transformers: $4\times$ more FLOPs for attention layers parameters and $\{1 + 1/2 + 1/4 + 1/8 = 1.875\}\times$ more FLOPs for MLP layers.

## E    Further Analysis on Language Decoders

### E.1    KL Divergence Between S, M, L and XL Models

Figure 8 showcases the smoother consistency calculation between two generative models measured with KL-divergence of the smaller model's outputs with the larger model outputs. Similar to the exact match style hard consistency metric used in the main paper, there is a significant gap between the consistency of MatLM's submodels with the MatLM-XL model and between that of the corresponding baseline models. This points to how sampling strategies based on the output probabilities do not change the behavioral consistency between two models and that it still follows the trend of generating the token with the highest probability. This smoother notion of consistency argues for the metric-space preservation given that the output classifier/embedding matrix is shared across all the submodels of MatLM.

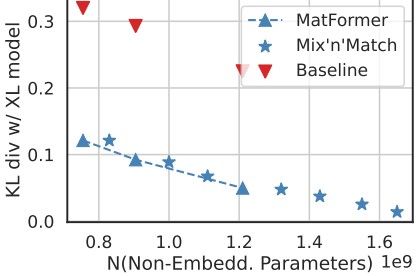

Figure 8: The smoother variant of consistency measures the KL divergence between the smaller models and the corresponding XL model. This metric, unlike the exact match accuracy variant, also accounts for different sampling strategies on the output distribution during deployment. In this figure, we plot KL divergence of S, M, L granularities with respect to XL for the 2.6B parameter model.

## E.2 Ablations on Training Method

We experiment with several aspects of our training method on a 850M parameter MatLM. Our training procedure is unique compared to others (further discussed in Section 2) in 2 ways: (a) we learn all granularities in the same weight space and (b) we use joint optimization as described in Section 3. To assess the effect of these differences on performance, first we train a Transformer model with independent FFN modules with {S, M, L, XL} granularites using joint optimization (Independent modules). Next, we train a MatLM model with the only difference being that at each step, we optimize for a single granularity chosen uniformly at random (Subsampling). We find that joint optimizing a MatLM performs significantly better than these baselines, implying efficacy of both aspects of our training method.

Table 5: We compared the validation loss of models from Joint Optimization to training MatLMs with independent MLP modules for each granularity (Independent modules) and sampling a single granularity to optimize for at each step (Subsampling) for 850M parameter models. We find that Joint Optimization performs significantly better than both these methods.

| Model | Training Strategy | XL | L | M | S |
|---|---|---|---|---|---|
| Baseline | - | 2.840 | 2.910 | 2.9710 | 3.017 |
| MatFormer | Joint Optimization | 2.874 | 2.928 | 2.980 | 3.030 |
| | Independent MLP modules | 2.894 | 2.942 | 2.985 | 3.030 |
| | Subsampling | 2.929 | 2.946 | 2.999 | 3.049 |

We discuss additional ablations such as re-weighting losses to improve the performance of the XL model in Appendix E.4, and additionally studied scaling trends for these ablations. We found the reweighting loss trick to be especially powerful, bringing the performance on downstream evals within $0.1\%$ for the XL model. This also nudges us towards finding better hyperparameters and weight initializations for reliable scaling of MatLMs (Yang et al., 2022).

## E.3 Changing Embedding Size

Because of the ubiquity of 64k vocabs size (Brown et al., 2020) we additionally train models upto 201M non-embedding parameters similar to those described in Appendix A, except that the embedding size is 64k (the largest model corresponds to the 463M parameter model). We plot the scaling trends in Figure 13. Though 4 models is not enough to extrapolate a trend, we observe that the scaling trend for validation loss appears to be similar.

## E.4 Reweighting Strategies

We additionally experiment with reweighting the losses for the individual granularities in order to boost the performance of the largest granularity while minimally impacting the performance of the smaller granularities. We present the relative weights used in Table 6 as $\lambda_4 : \lambda_3 : \lambda_2 : \lambda_1$, and find that in general, upweighting the largest granularity greatly improves quality. Another interesting related direction for improving MatFormer performance further is granularity appropriate initialization (Yang et al., 2022).

## E.5 Scaling Laws for Reweighted Strategy

We conduct scaling experiments similar to those described in Section 4.1 for the reweighed models, specifically for models with the ratio $2 : 1.5 : 1.25 : 1$, and plot the results in Figure 14. We note that the scaling trend is similar to the MatLM with a $1 : 1 : 1 : 1$ relative weighting ($a = 19.889, b = -0.130, c = 1.374$), but with a slightly better validation loss .

# F Further Analysis on Vision Encoders

## F.1 Decoupling Effect of MatFormer on Pretraining and Finetuning

Table 7 investigates the effect of MatFormer on pretaining and finetuning phases of ViT-L/16 model. ViT-L/16 is typically pretrained on ImageNet-21K and then finetuned on ImageNet-1K for the final

Table 6: For 850M model, we experiment with modifying $\mathcal{L}_{\text{JOINT}}$ to use a weighted average as opposed to an unweighted average, and report the results across all granularities. We find that all strategies that upweight the loss for the largest granularity perform well, with modest degradation on the M and S granularties.

| Model | Relative Weights | XL | L | M | S |
|---|---|---|---|---|---|
| Baseline | N/A | 2.840 | 2.910 | 2.971 | 3.017 |
| MatFormer | 1:1:1:1 | 2.874 | 2.928 | 2.980 | 3.030 |
| | $2:1.5:1.25:1$ | 2.867 | 2.927 | 2.986 | 3.051 |
| | $1:1.25:1.5:2$ | 2.883 | 2.936 | 2.982 | 3.026 |
| | $2:1:1:1$ | 2.863 | 2.929 | 2.985 | 3.043 |
| | $\sqrt{8}:\sqrt{4}:\sqrt{2}:1$ | 2.862 | 2.924 | 2.990 | 3.063 |

evaluation. Table 7 shows that having a MatFormer during pretraining generates a better model for downstream finetuning compared to regular ViT pertaining. At the same time, finetuning a vanilla pretrained ViT with MatFormer results in flexibility being induced into the model. Despite being up to 2% less accurate than its counterparts at some granularities, a fine-tuned MatViT learned to reallocate the information to provide strong nested models. Considering that this is insignificant compared to pretaining costs, possible to take the largest pretrained ViT model and finetune with MatFormer to obtain a deployable MatViT variant.

Table 7: $2 \times 2$ grid of pairs to evaluate (top-1 accuracy (%)) the effects of MatFormer and standard training on the pretraining (PT) on ImageNet-21K and finetuning (FT) on ImageNet-1K using a L/16 architecture. Using a MatFormer during pretraining helps bring more accurate, and elastic encoders for downstream uses.

| PT↓ / FT→ | # Params (M) | ViT | MatViT |
|---|---|---|---|
| ViT | 306 | 85.26 | 85.57 |
| | 206 | 85.12 | 84.27 |
| | 156 | 85.02 | 82.79 |
| | 131 | 84.42 | 82.1 |
| MatViT | 306 | 85.58 | 85.61 |
| | 206 | – | 85.40 |
| | 156 | – | 85.02 |
| | 131 | – | 84.41 |

## F.2 Traditional Image Retrieval Evaluation

Table 8 showcases traditional image retrieval evaluation on ImageNet-1K where the query and the document encoders are the same for nearest neighbor retrieval. The 1-nearest neighbor (NN) based evaluation closely follows one-vs-all classification results shown in Figure 4. Both MatViT variants B/16 and L/16 have submodels that have as good or better retrieval performance compared to their independently trained counterparts. Concretely, MatViT-based retrieval can be up to 0.5% more accurate than the baselines while a 200M parameter MatViT submodel can be more accurate than the 300M parameter ViT baseline.

Table 8: Image retrieval 1-NN accuracy (%) when the query and document encoders are the same model. Similar to the image classification results, MatViT variants either match or outperform the corresponding standard ViT counterparts. Note that all the smaller models of a given model in MatViT are extracted for free while the baselines have to be explicitly trained for the constraints.

| Encoder | # Params (M) | ViT | MatViT |
|---------|--------------|-------|--------|
| B/16    | 85           | 77.46 | 77.38  |
|         | 57           | 76.58 | 76.41  |
|         | 43           | 74.90 | 74.49  |
|         | 36           | 71.44 | 71.72  |
| L/16    | 300          | 83.17 | 83.67  |
|         | 200          | 82.92 | 83.23  |
|         | 150          | 82.81 | 82.89  |
|         | 125          | 82.22 | 82.14  |

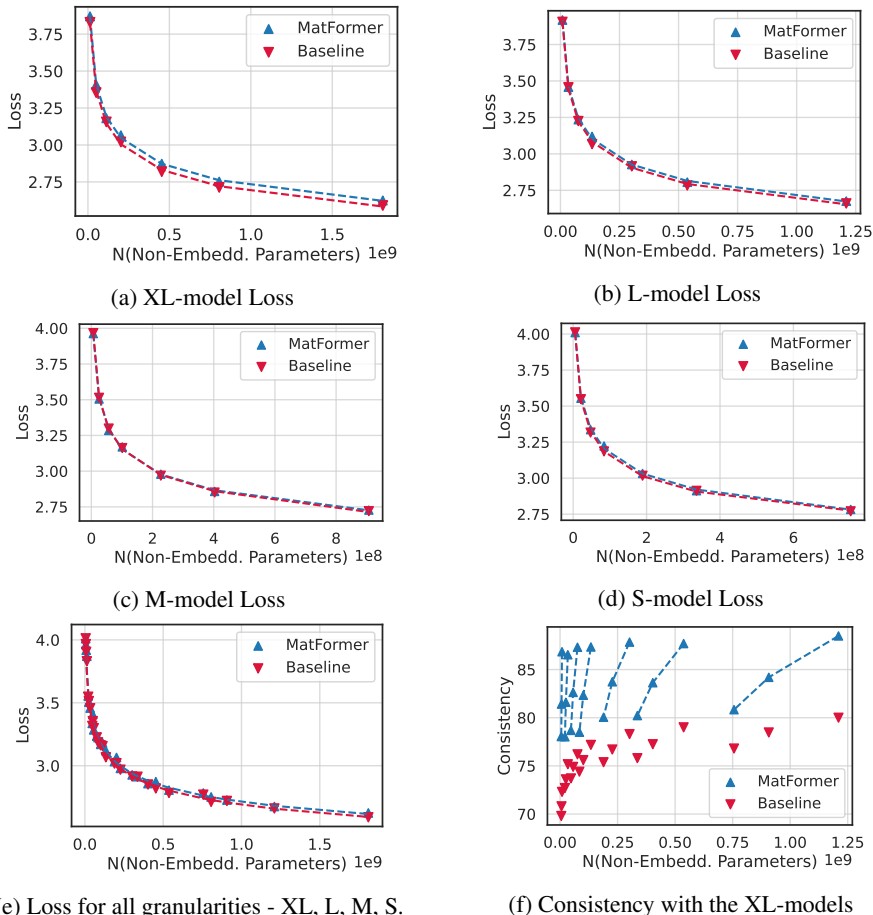

(a) XL-model Loss

(b) L-model Loss

(c) M-model Loss

(d) S-model Loss

(e) Loss for all granularities - XL, L, M, S.

(f) Consistency with the XL-models

Figure 9: We train various decoder-only MatLM models at a range of sizes from 78M to 2.6B parameters and observe the scaling trends for each model granularity on validation loss. We observe that the gap between MatLM and the baseline appears to be constant at each granularity. The consistency between the submodels of granularities and the XL models shows the effect of MatFormer joint training on natively ensuring similar behavior across submodels.

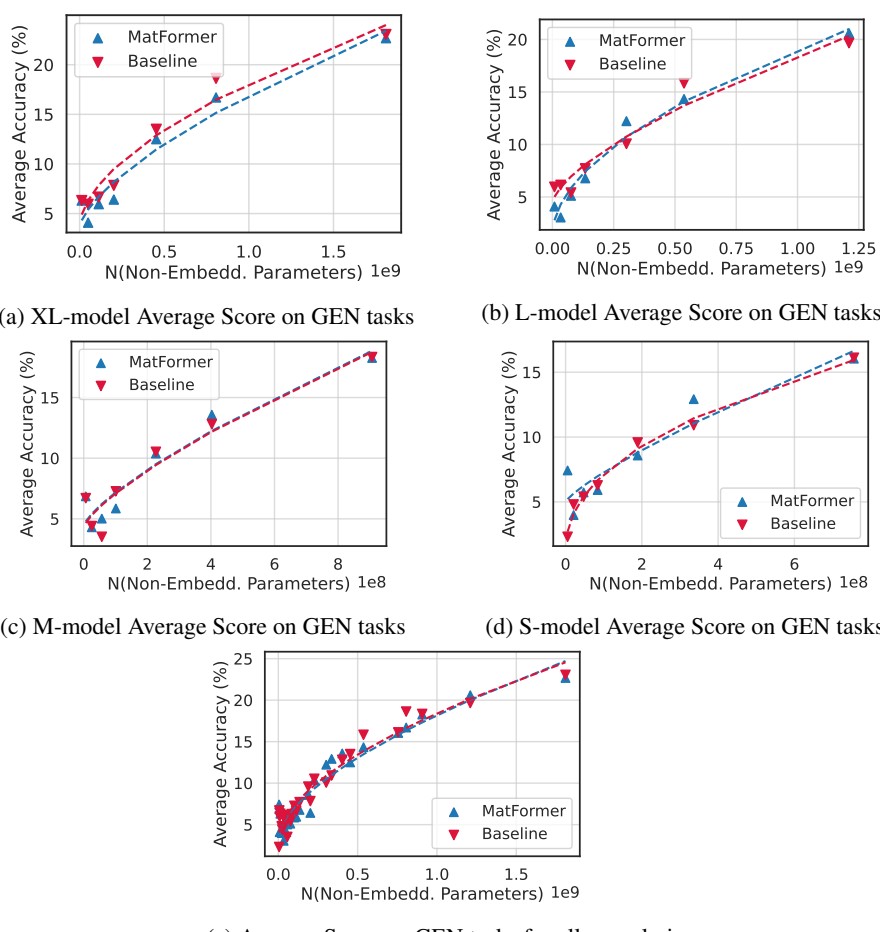

(a) XL-model Average Score on GEN tasks

(b) L-model Average Score on GEN tasks

(c) M-model Average Score on GEN tasks

(d) S-model Average Score on GEN tasks

(e) Average Score on GEN tasks for all granularities - XL, L, M, S.

Figure 10: We train various decoder-only MatLM models at a range of sizes from 78M to 2.6B parameters and observe the scaling trends for each model granularity for the average score on GEN tasks 1-shot evaluation. We observe that the gap between MatLM and the baseline reduces with scale, outperforming the baselines for S, M, L granularities for the largest models.

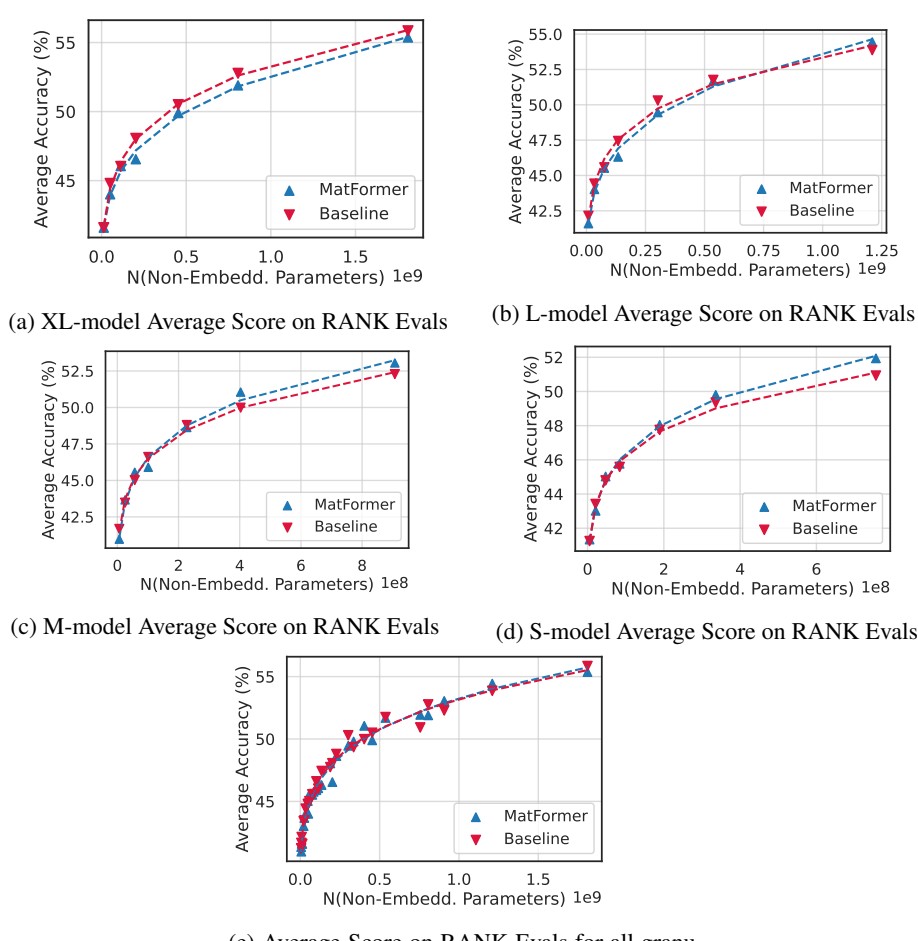

(a) XL-model Average Score on RANK Evals

(b) L-model Average Score on RANK Evals

(c) M-model Average Score on RANK Evals

(d) S-model Average Score on RANK Evals

(e) Average Score on RANK Evals for all granu-
larities - XL, L, M, S

Figure 11: We train various decoder-only MatLM models at a range of sizes from 78M to 2.6B parameters and observe the scaling trends for each model granularity for the average score on RANK 1-shot evaluation. We observe that the gap between MatLM and the baseline reduces with scale, outperforming the baselines for S, M, L granularities for the largest models.

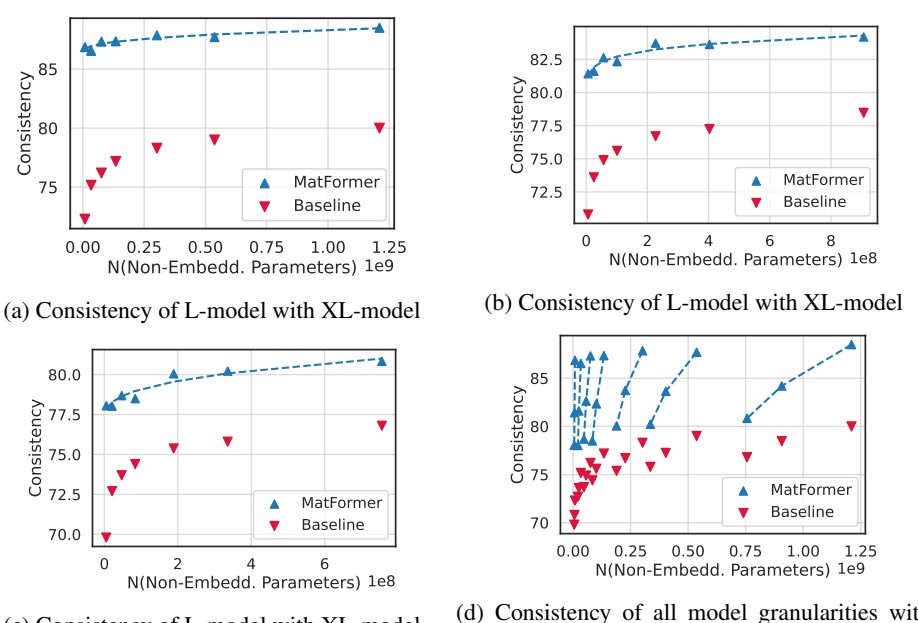

(a) Consistency of L-model with XL-model

(b) Consistency of L-model with XL-model

(c) Consistency of L-model with XL-model

(d) Consistency of all model granularities with XL-model - L, M, S

Figure 12: We train various decoder-only MatLM models at a range of sizes from 78M to 2.6B parameters and observe the scaling trends for each submodel S, M, L for the consistency with the XL model. We observe that the gap between MatLM and the baseline reduces with scale, but one would need to scale the baseline by many orders of magnitude to have consistency comparable to that of MatLMs.

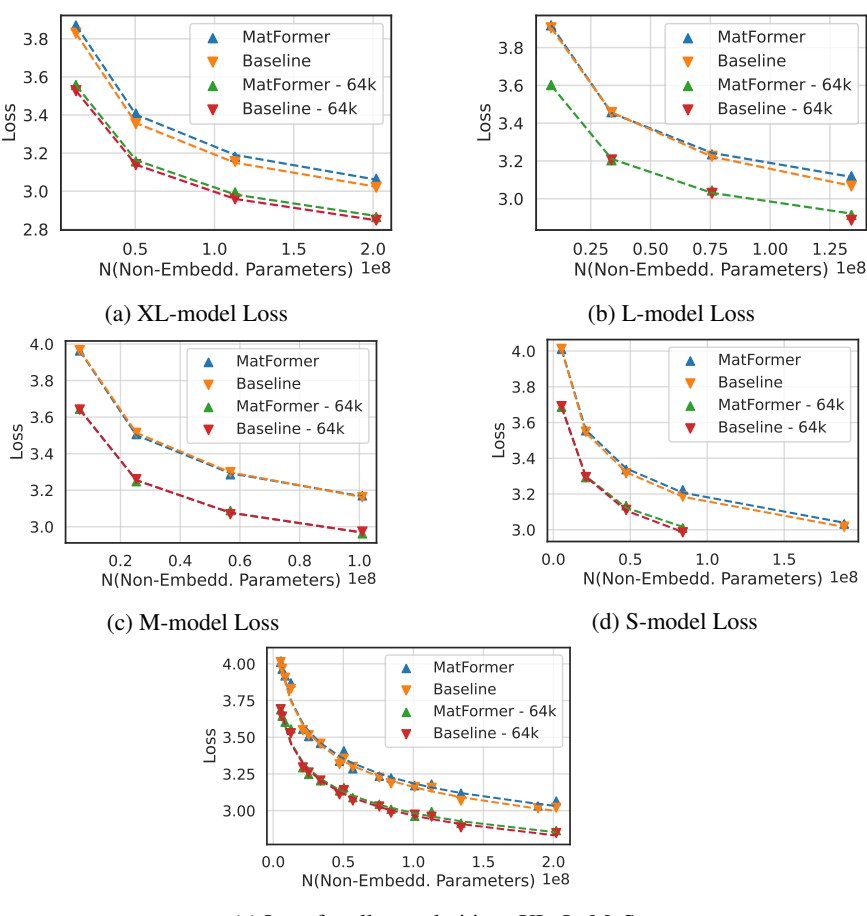

(a) XL-model Loss

(b) L-model Loss

(c) M-model Loss

(d) S-model Loss

(e) Loss for all granularities - XL, L, M, S

Figure 13: We train various decoder-only MatLM models at a range of sizes from 29M to 267M parameters with an embedding size of 64k and observe the scaling trends for each model granularity on validation loss. We observe that the gap between MatLM and the baseline appears to be constant at each granularity, similar to what is observed in Figure 9.

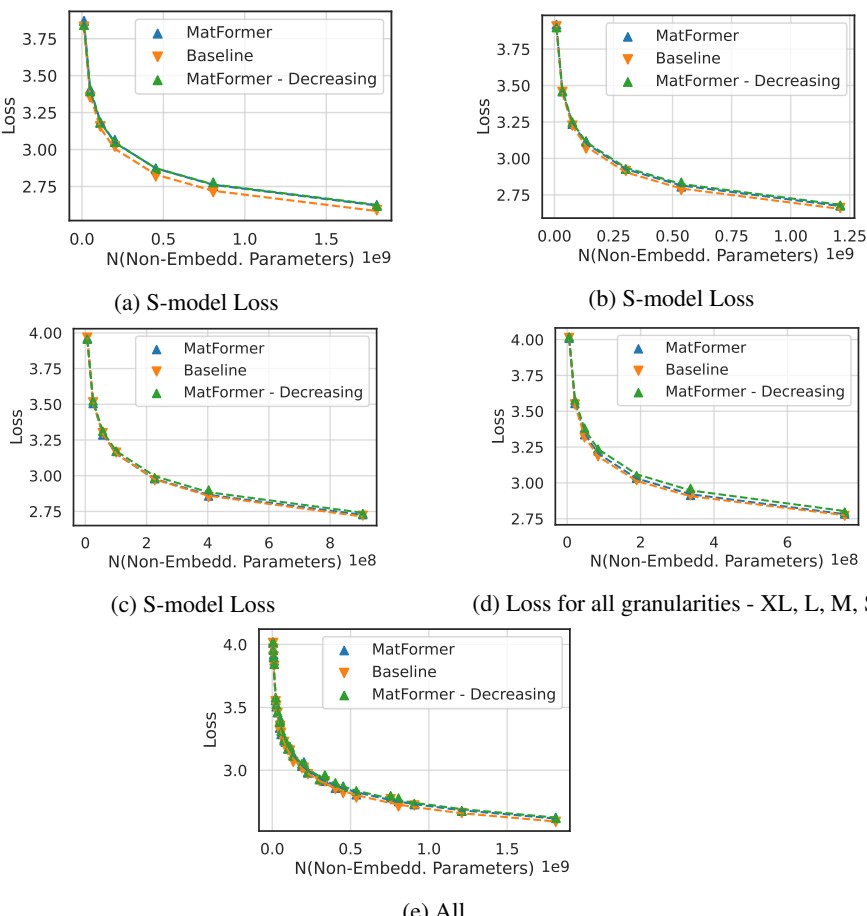

(a) S-model Loss

(b) S-model Loss

(c) S-model Loss

(d) Loss for all granularities - XL, L, M, S

(e) All

Figure 14: We train various decoder-only MatLM models at a range of sizes from 78M to 2.6B parameters with a reweighing ratio of $2 : 1.5 : 1.25 : 1$ and observe the scaling trends for each model granularity on validation loss. We observe that the gap between MatLM and the baseline appears to be constant at each granularity, similar to what is observed in Figure 9.

Table 9: Downstream Eval numbers and development set log perplexity loss on 78M model size granularities.

| Downstream Task | Baseline-S | MatLM-S | Baseline-M | MatLM-M | Baseline-L | MatLM-L | Baseline-XL | MatLM-XL |
|---|---|---|---|---|---|---|---|---|
| TriviaQA (EM) | 0.14 | 0.16 | 0.19 | 0.25 | 0.14 | 0.3 | 0.19 | 0.28 |
| NaturalQuestions (EM) | 0.06 | 0.03 | 0.03 | 0.06 | 0.03 | 0.03 | 0.03 | 0.03 |
| WebQuestions (EM) | 0.1 | 0.2 | 0.15 | 0.2 | 0.2 | 0.3 | 0.3 | 0.3 |
| LAMBADA | 0.06 | 0.02 | 0.02 | 0 | 0.02 | 0 | 0 | 0 |
| HellaSwag | 25.42 | 26.28 | 26 | 25.87 | 25.95 | 25.9 | 25.95 | 25.94 |
| StoryCloze | 52.81 | 53.39 | 53.13 | 53.34 | 54.46 | 53.5 | 54.46 | 54.36 |
| WSC | 52.98 | 51.93 | 53.68 | 50.88 | 55.79 | 54.04 | 52.28 | 52.63 |
| WinoGrande | 48.46 | 51.54 | 51.54 | 47.99 | 50.99 | 48.46 | 48.86 | 49.41 |
| Winograd | 53.11 | 52.75 | 52.38 | 53.85 | 55.31 | 55.31 | 52.75 | 55.68 |
| SQuAD v2 (EM) | 11.19 | 36.71 | 33.14 | 33.77 | 20.08 | 29.17 | 22.78 | 30.97 |
| RACE-H | 25.53 | 25.84 | 24.73 | 25.44 | 26.07 | 25.9 | 25.96 | 25.84 |
| RACE-M | 29.18 | 30.15 | 28.83 | 29.94 | 28.83 | 30.43 | 29.74 | 31.48 |
| PIQA | 55.77 | 55.22 | 54.62 | 55.28 | 54.52 | 54.79 | 56.86 | 54.08 |
| ARC-C | 21.5 | 20.9 | 21.08 | 21.67 | 21.59 | 21.33 | 22.35 | 22.1 |
| ARC-E | 34.55 | 35.48 | 34.3 | 35.73 | 34.89 | 36.11 | 34.55 | 35.98 |
| OpenBookQA | 25.4 | 28.6 | 27.6 | 28 | 28.2 | 28 | 29.8 | 29 |
| BoolQ | 48.72 | 44.89 | 51.87 | 47.37 | 51.28 | 46.85 | 52.11 | 45.87 |
| COPA | 62 | 64 | 62 | 61 | 63 | 63 | 60 | 60 |
| RTE | 53.79 | 52.35 | 52.35 | 51.99 | 51.26 | 54.51 | 51.99 | 52.71 |
| WiC | 49.53 | 47.34 | 49.06 | 47.34 | 47.34 | 47.34 | 47.65 | 47.34 |
| MultiRC (F1) | 53.17 | 51.72 | 53.42 | 53.28 | 56.86 | 53.82 | 55.46 | 53.42 |
| ReCoRD | 39.52 | 39.22 | 40.03 | 39.95 | 40.55 | 40.42 | 40.8 | 40.83 |
| CB | 41.07 | 42.86 | 44.64 | 39.29 | 44.64 | 41.07 | 42.86 | 44.64 |
| ANLI-R1 | 30.9 | 32 | 32.3 | 31.9 | 32.5 | 32.3 | 32.5 | 31.7 |
| ANLI-R2 | 31.1 | 30.9 | 31.1 | 30.1 | 30.7 | 30.8 | 30.6 | 30.3 |
| ANLI-R3 | 31.75 | 30.75 | 30.58 | 30.25 | 30.33 | 29.67 | 30 | 30.17 |
| Average | 33.76 | 34.82 | 34.95 | 34.41 | 34.83 | 34.74 | 34.65 | 34.81 |
| Avg over GEN Taks | 2.31 | 7.42 | 6.7 | 6.85 | 4.09 | 5.96 | 4.66 | 6.31 |
| Avg over RANK Tasks | 41.25 | 41.34 | 41.68 | 40.97 | 42.15 | 41.6 | 41.79 | 41.59 |
| Dev set log pplx | 4.010 | 4.012 | 3.97 | 3.96 | 3.905 | 3.908 | 3.83 | 3.868 |

Table 10: Downstream Eval numbers and development set log perplexity loss on 180M model size granularities.

| Downstream Task | Baseline-S | MatLM-S | Baseline-M | MatLM-M | Baseline-L | MatLM-L | Baseline-XL | MatLM-XL |
|---|---|---|---|---|---|---|---|---|
| TriviaQA (EM) | 1.04 | 0.9 | 0.98 | 1.26 | 1.16 | 1.89 | 1.86 | 2.00 |
| NaturalQuestions (EM) | 0.08 | 0.11 | 0.14 | 0.08 | 0.3 | 0.11 | 0.28 | 0.11 |
| WebQuestions (EM) | 0.59 | 0.94 | 0.44 | 0.98 | 1.28 | 0.89 | 1.33 | 0.79 |
| LAMBADA | 0.16 | 0.68 | 0.43 | 1.16 | 1.51 | 0.95 | 0.49 | 0.99 |
| HellaSwag | 27.77 | 27.3 | 27.45 | 27.61 | 27.58 | 27.84 | 28.86 | 28.56 |
| StoryCloze | 56.33 | 56.07 | 57.03 | 56.87 | 57.3 | 57.78 | 58.63 | 58.52 |
| WSC | 55.44 | 55.44 | 56.49 | 60.35 | 58.25 | 58.6 | 57.54 | 58.6 |
| WinoGrande | 52.01 | 50.12 | 50.28 | 49.17 | 51.22 | 50.43 | 51.54 | 49.09 |
| Winograd | 54.21 | 55.68 | 56.78 | 57.51 | 61.54 | 58.61 | 60.44 | 61.17 |
| SQuAD v2 (EM) | 22.13 | 17.28 | 20.05 | 18.02 | 26.42 | 11.42 | 25.76 | 16.53 |
| RACE-H | 27.93 | 27.9 | 27.5 | 28.53 | 28.7 | 28.82 | 28.73 | 28.73 |
| RACE-M | 33.29 | 34.47 | 34.19 | 34.05 | 34.54 | 33.91 | 33.29 | 34.19 |
| PIQA | 57.13 | 58.05 | 56.91 | 57.94 | 57.94 | 58.00 | 59.52 | 58.92 |
| ARC-C | 22.53 | 22.61 | 23.63 | 22.27 | 24.06 | 22.1 | 24.66 | 23.55 |
| ARC-E | 40.24 | 39.39 | 40.19 | 40.49 | 41.71 | 40.74 | 41.62 | 41.16 |
| OpenBookQA | 30.60 | 31.00 | 30.80 | 31.80 | 31.00 | 32.80 | 34.00 | 32.6 |
| BoolQ | 54.13 | 52.23 | 52.45 | 52.05 | 55.63 | 52.17 | 55.9 | 48.44 |
| COPA | 62 | 61 | 61 | 61 | 61 | 64 | 64 | 65 |
| RTE | 52.71 | 53.07 | 52.35 | 53.43 | 50.54 | 52.71 | 52.71 | 52.71 |
| WiC | 47.34 | 51.41 | 47.34 | 49.37 | 47.96 | 47.81 | 47.65 | 47.34 |
| MultiRC (F1) | 54.34 | 53.34 | 45.65 | 56.12 | 47.47 | 52.62 | 47.62 | |
| ReCoRD | 48.58 | 49.4 | 48.99 | 50.13 | 50.56 | 51.25 | 52.82 | 52.51 |
| CB | 42.86 | 44.64 | 42.86 | 44.64 | 39.29 | 44.64 | 42.86 | 42.86 |
| ANLI-R1 | 31.8 | 32.6 | 31.8 | 32.4 | 32.4 | 32.8 | 32.2 | 32.1 |
| ANLI-R2 | 30.5 | 29.8 | 31.1 | 29.8 | 32.00 | 30.5 | 30.5 | 30.1 |
| ANLI-R3 | 30.08 | 30.25 | 30.5 | 32.00 | 33.5 | 31.42 | 30.67 | 30.42 |
| Average | 35.99 | 35.51 | 35.96 | 36.1 | 37.06 | 36.14 | 37.33 | 36.33 |
| GPT3-GEN | 4.8 | 3.98 | 4.41 | 4.3 | 6.14 | 3.05 | 5.94 | 4.08 |
| GPT3-RANK | 43.42 | 43.02 | 43.48 | 43.67 | 44.42 | 44.02 | 44.8 | 44.01 |
| Dev set log pplx | 3.55 | 3.55 | 3.512 | 3.505 | 3.456 | 3.458 | 3.354 | 3.40 |

Table 11: Downstream Eval numbers and development set log perplexity loss on 310M model size granularities.

| Downstream Task | Baseline-S | MatLM-S | Baseline-M | MatLM-M | Baseline-L | MatLM-L | Baseline-XL | MatLM-XL |
|---|---|---|---|---|---|---|---|---|
| TriviaQA (EM) | 2.09 | 2.4 | 2.2 | 3.17 | 2.84 | 2.73 | 5.18 | 3.12 |
| NaturalQuestions (EM) | 0.11 | 0.28 | 0.28 | 0.5 | 0.58 | 0.3 | 0.91 | 0.61 |
| WebQuestions (EM) | 2.12 | 1.38 | 1.08 | 1.67 | 1.67 | 1.43 | 2.41 | 1.57 |
| LAMBADA | 0.29 | 1.79 | 0.66 | 1.92 | 1.9 | 2.46 | 2.76 | 2.64 |
| HellaSwag | 29.89 | 29.69 | 30.05 | 30.02 | 31.18 | 30.63 | 32.52 | 31.58 |
| StoryCloze | 59.17 | 58.85 | 59.54 | 60.13 | 60.24 | 60.5 | 61.68 | 61.36 |
| WSC | 61.05 | 59.65 | 59.3 | 58.6 | 61.75 | 56.84 | 58.95 | 57.19 |
| WinoGrande | 51.46 | 52.88 | 49.57 | 50.91 | 52.41 | 50.75 | 50.91 | 52.01 |
| Winograd | 55.68 | 56.04 | 57.88 | 59.71 | 63 | 59.71 | 61.17 | 60.07 |
| SQuAD v2 (EM) | 22.38 | 22.79 | 13.38 | 17.83 | 20.03 | 18.66 | 22.03 | 21.81 |
| RACE-H | 29.45 | 28.33 | 28.9 | 28.67 | 29.22 | 29.07 | 29.67 | 28.79 |
| RACE-M | 35.31 | 36.14 | 36.14 | 36.91 | 36.42 | 36.14 | 37.6 | 36.07 |
| PIQA | 58.98 | 59.9 | 59.58 | 59.85 | 59.79 | 60.45 | 62.19 | 60.61 |
| ARC-C | 23.38 | 20.82 | 23.21 | 21.33 | 23.81 | 23.21 | 25 | 22.95 |
| ARC-E | 42.3 | 42.34 | 44.11 | 43.52 | 44.53 | 44.44 | 46.8 | 45.62 |
| OpenBookQA | 32.8 | 35.2 | 34.6 | 36.4 | 35.2 | 35.8 | 36.8 | 36.6 |
| BoolQ | 53.43 | 59.05 | 55.32 | 58.72 | 52.87 | 57.22 | 54.22 | 55.6 |
| COPA | 61 | 61 | 61 | 66 | 64 | 63 | 60 | 66 |
| RTE | 52.71 | 54.51 | 53.43 | 51.62 | 51.62 | 53.07 | 54.15 | 49.46 |
| WiC | 47.18 | 48.43 | 47.65 | 49.22 | 47.65 | 50.16 | 47.34 | 51.25 |
| MultiRC (F1) | 53.07 | 51.69 | 53.5 | 51.36 | 48.46 | 47.14 | 45.72 | 46.23 |
| ReCoRD | 54.34 | 53.86 | 55.18 | 55.33 | 56.75 | 56.79 | 58.39 | 58.07 |
| CB | 42.86 | 46.43 | 42.86 | 46.43 | 42.86 | 46.43 | 50 | 51.79 |
| ANLI-R1 | 32 | 31.3 | 32 | 32.2 | 32.5 | 32.3 | 32.2 | 32.8 |
| ANLI-R2 | 32.6 | 30.2 | 30.9 | 29.8 | 30.6 | 31.2 | 29.8 | 30.9 |
| ANLI-R3 | 32.08 | 29.25 | 30.75 | 30.08 | 32.17 | 31.25 | 31.5 | 32.17 |
| Average | 37.22 | 37.47 | 37.04 | 37.77 | 37.85 | 37.76 | 38.46 | 38.34 |
| Avg over GEN Taks | 5.4 | 5.73 | 3.52 | 5.02 | 5.41 | 5.12 | 6.66 | 5.95 |
| Avg over RANK Tasks | 44.8 | 45.03 | 45.02 | 45.56 | 45.57 | 45.53 | 46.03 | 46.05 |
| Dev set log pplx | 3.31 | 3.33 | 3.30 | 3.285 | 3.224 | 3.235 | 3.15 | 3.18 |

Table 12: Downstream Eval numbers and development set log perplexity loss on 463M model size granularities.

| Downstream Task | Baseline-S | MatLM-S | Baseline-M | MatLM-M | Baseline-L | MatLM-L | Baseline-XL | MatLM-XL |
|---|---|---|---|---|---|---|---|---|
| TriviaQA (EM) | 4.63 | 3.87 | 4.87 | 4.55 | 6.11 | 5.63 | 8.09 | 6.48 |
| NaturalQuestions (EM) | 0.61 | 0.58 | 0.8 | 0.89 | 0.94 | 1.16 | 1.66 | 1.25 |
| WebQuestions (EM) | 2.31 | 1.62 | 2.26 | 2.02 | 2.85 | 2.31 | 2.85 | 2.56 |
| LAMBADA | 2.1 | 1.65 | 2.6 | 2.1 | 3.94 | 2.93 | 3.49 | 3.49 |
| HellaSwag | 32.12 | 31.57 | 32.83 | 32.16 | 33.8 | 33.48 | 36.21 | 35.08 |
| StoryCloze | 61.25 | 60.98 | 61.36 | 61.46 | 63.66 | 62.21 | 64.24 | 64.08 |
| WSC | 57.54 | 64.91 | 61.4 | 62.11 | 66.32 | 62.11 | 61.05 | 63.16 |
| WinoGrande | 52.33 | 51.38 | 49.09 | 50.99 | 52.64 | 50.36 | 53.12 | 52.64 |
| Winograd | 60.07 | 63.74 | 60.07 | 62.27 | 67.4 | 61.54 | 68.5 | 63.74 |
| SQuAD v2 (EM) | 21.7 | 21.85 | 25.8 | 19.71 | 24.69 | 21.85 | 23.08 | 18.28 |
| RACE-H | 29.85 | 29.45 | 29.47 | 29.79 | 30.56 | 29.79 | 30.7 | 30.02 |
| RACE-M | 37.53 | 37.6 | 37.33 | 38.93 | 40.39 | 39.62 | 40.95 | 39.21 |
| PIQA | 61.26 | 61.53 | 61.48 | 62.08 | 60.99 | 63.22 | 63.17 | 63.71 |
| ARC-C | 23.04 | 22.7 | 24.06 | 22.35 | 24.49 | 22.18 | 23.72 | 23.63 |
| ARC-E | 45.83 | 44.44 | 46.3 | 45.62 | 47.73 | 47.85 | 51.73 | 49.12 |
| OpenBookQA | 37.2 | 36.4 | 37 | 37.8 | 36.4 | 39.2 | 41 | 38.4 |
| BoolQ | 52.39 | 52.69 | 56.12 | 52.05 | 50.28 | 51.28 | 54.98 | 47.95 |
| COPA | 67 | 62 | 73 | 63 | 71 | 63 | 67 | 66 |
| RTE | 52.35 | 53.07 | 53.43 | 52.71 | 52.35 | 52.71 | 52.35 | 51.99 |
| WiC | 47.34 | 47.34 | 47.34 | 47.34 | 47.34 | 47.34 | 47.34 | 47.34 |
| MultiRC (F1) | 45.63 | 46.02 | 54.4 | 46.38 | 52.79 | 49.28 | 52.34 | 41.71 |
| ReCoRD | 57.58 | 58.65 | 59.31 | 59.71 | 60.87 | 61 | 63.42 | 61.77 |
| CB | 42.86 | 42.86 | 44.64 | 42.86 | 44.64 | 42.86 | 42.86 | 42.86 |
| ANLI-R1 | 32.6 | 32.5 | 31.7 | 33.1 | 31.4 | 32.3 | 32.5 | 32.6 |
| ANLI-R2 | 30.7 | 30.7 | 28.4 | 30.5 | 30.4 | 30.6 | 31.2 | 31.8 |
| ANLI-R3 | 30.83 | 30.67 | 30.08 | 30.75 | 30.83 | 30.67 | 30.92 | 30.75 |
| Average | 38.02 | 38.11 | 39.04 | 38.2 | 39.8 | 38.71 | 40.33 | 38.83 |
| Avg over GEN Taks | 6.27 | 5.91 | 7.27 | 5.85 | 7.71 | 6.78 | 7.84 | 6.41 |
| Avg over RANK Tasks | 45.59 | 45.77 | 46.61 | 45.9 | 47.44 | 46.31 | 48.06 | 46.55 |
| Dev set log pplx | 3.205 | 3.217 | 3.16 | 3.16 | 3.096 | 3.11 | 3.023 | 3.06 |

Table 13: Downstream Eval numbers and development set log perplexity loss on 850M model size granularities.

| Downstream Task | Baseline-S | MatLM-S | Baseline-M | MatLM-M | Baseline-L | MatLM-L | Baseline-XL | MatLM-XL |
|---|---|---|---|---|---|---|---|---|
| TriviaQA (EM) | 9.26 | 6.62 | 10.82 | 9.78 | 11.07 | 11.72 | 13.31 | 13.76 |
| NaturalQuestions (EM) | 1.66 | 0.89 | 1.69 | 1.58 | 2.24 | 2.38 | 2.66 | 2.74 |
| WebQuestions (EM) | 3.89 | 3.35 | 4.08 | 4.18 | 3.74 | 4.43 | 4.08 | 5.31 |
| LAMBADA | 3.2 | 8.25 | 6.97 | 10.83 | 8.19 | 10.44 | 14.03 | 10.83 |
| HellaSwag | 36.11 | 36.64 | 38.26 | 37.7 | 40.63 | 39.64 | 43.4 | 42.55 |
| StoryCloze | 64.78 | 65.26 | 66.33 | 66.17 | 68.25 | 67.13 | 71.25 | 69.64 |
| WSC | 66.32 | 65.96 | 63.16 | 64.21 | 69.82 | 69.12 | 70.53 | 68.42 |
| WinoGrande | 52.17 | 51.54 | 52.25 | 52.57 | 55.17 | 52.96 | 54.14 | 54.62 |
| Winograd | 68.13 | 69.23 | 67.03 | 71.43 | 71.06 | 70.33 | 72.16 | 72.89 |
| SQuAD v2 (EM) | 29.9 | 23.79 | 29.07 | 25.51 | 25.07 | 26.39 | 33.41 | 28.46 |
| RACE-H | 30.39 | 30.76 | 31.93 | 31.88 | 32.53 | 31.88 | 33.79 | 32.73 |
| RACE-M | 40.95 | 40.95 | 42.06 | 41.16 | 42.27 | 42.55 | 44.64 | 42.48 |
| PIQA | 64.04 | 63.98 | 64.64 | 64.91 | 65.45 | 65.23 | 67.25 | 66.21 |
| ARC-C | 24.49 | 24.15 | 26.71 | 24.91 | 26.71 | 26.54 | 27.13 | 27.47 |
| ARC-E | 52.15 | 51.01 | 53.66 | 52.95 | 56.27 | 54.92 | 57.11 | 56.57 |
| OpenBookQA | 38.2 | 40.4 | 40.8 | 41.2 | 42.8 | 40.8 | 43 | 42 |
| BoolQ | 52.63 | 50.31 | 51.9 | 47.8 | 56.73 | 50.15 | 55.6 | 48.41 |
| COPA | 68 | 73 | 68 | 73 | 71 | 73 | 73 | 76 |
| RTE | 51.62 | 51.99 | 52.71 | 52.35 | 51.62 | 51.99 | 53.07 | 52.71 |
| WiC | 47.34 | 47.18 | 47.34 | 47.18 | 47.34 | 47.18 | 47.34 | 47.18 |
| MultiRC (F1) | 44.37 | 51.32 | 52.11 | 50.46 | 54.7 | 53 | 37.58 | 47.16 |
| ReCoRD | 63.52 | 64.27 | 65.03 | 65.36 | 67.55 | 66.53 | 69.56 | 68.03 |
| CB | 42.86 | 37.5 | 42.86 | 42.86 | 42.86 | 42.86 | 46.43 | 39.29 |
| ANLI-R1 | 30.9 | 31.8 | 33.7 | 32.1 | 31.7 | 32.2 | 32.6 | 32.4 |
| ANLI-R2 | 31.8 | 31.5 | 31.5 | 30.9 | 31.1 | 30.6 | 30.4 | 30.8 |
| ANLI-R3 | 32 | 30.25 | 32.83 | 30.17 | 30.75 | 30 | 30.58 | 30.25 |
| Average | 40.41 | 40.46 | 41.44 | 41.27 | 42.56 | 42.08 | 43.39 | 42.65 |
| Avg over GEN Taks | 9.58 | 8.58 | 10.53 | 10.38 | 10.06 | 11.07 | 13.5 | 12.22 |
| Avg over RANK Tasks | 47.75 | 48.05 | 48.8 | 48.63 | 50.3 | 49.46 | 50.5 | 49.9 |
| Dev set log pplx | 3.017 | 3.03 | 2.971 | 2.98 | 2.91 | 2.928 | 2.84 | 2.874 |

Table 14: Downstream Eval numbers and development set log perplexity loss on 1.3B model size granularities.

| Downstream Task | Baseline-S | MatLM-S | Baseline-M | MatLM-M | Baseline-L | MatLM-L | Baseline-XL | MatLM-XL |
|---|---|---|---|---|---|---|---|---|
| TriviaQA (EM) | 11.92 | 12 | 14.68 | 13.09 | 16.48 | 14.91 | 20.14 | 17.62 |
| NaturalQuestions (EM) | 1.88 | 2.19 | 2.24 | 2.47 | 3.07 | 2.99 | 4.79 | 4.13 |
| WebQuestions (EM) | 3.84 | 5.02 | 4.72 | 5.36 | 5.07 | 5.76 | 6.05 | 6.15 |
| LAMBADA | 7.3 | 9.94 | 13.55 | 12.34 | 17.97 | 13.51 | 22.65 | 19.21 |
| HellaSwag | 40.53 | 40.35 | 42.86 | 42.5 | 46 | 44.48 | 49.78 | 47.69 |
| StoryCloze | 67.29 | 68.2 | 69.75 | 69.91 | 72.37 | 71.14 | 73.81 | 72.8 |
| WSC | 64.56 | 65.96 | 64.91 | 69.12 | 67.72 | 69.82 | 72.63 | 69.82 |
| WinoGrande | 55.8 | 53.99 | 56.67 | 55.25 | 56.12 | 57.7 | 58.25 | 58.41 |
| Winograd | 71.06 | 68.5 | 67.77 | 70.7 | 73.99 | 70.33 | 72.53 | 72.89 |
| SQuAD v2 (EM) | 29.63 | 35.47 | 28.85 | 34.64 | 36.55 | 34.47 | 39.48 | 36.39 |
| RACE-H | 32.19 | 33.19 | 33.08 | 34.39 | 34.48 | 35.11 | 36.59 | 35.25 |
| RACE-M | 43.8 | 44.22 | 44.22 | 45.96 | 47.7 | 45.75 | 50.07 | 46.59 |
| PIQA | 66.49 | 64.36 | 66.05 | 66.38 | 67.52 | 66.97 | 69.1 | 67.68 |
| ARC-C | 27.99 | 25.77 | 27.65 | 27.22 | 29.01 | 28.75 | 30.55 | 31.48 |
| ARC-E | 56.44 | 54.08 | 58.54 | 57.03 | 59.85 | 58.84 | 63.26 | 61.83 |
| OpenBookQA | 41.4 | 42.2 | 41 | 42 | 43.4 | 42.8 | 44.8 | 45.4 |
| BoolQ | 52.57 | 49.85 | 54.86 | 52.42 | 53.76 | 56.06 | 55.35 | 53.52 |
| COPA | 70 | 75 | 69 | 77 | 74 | 74 | 77 | 75 |
| RTE | 52.35 | 53.07 | 53.07 | 52.35 | 54.15 | 53.43 | 52.35 | 49.82 |
| WiC | 47.34 | 47.34 | 47.18 | 47.34 | 47.34 | 47.34 | 48.43 | 47.02 |
| MultiRC (F1) | 42.98 | 46.69 | 43.82 | 49.09 | 45.29 | 48.2 | 40.99 | 46.42 |
| ReCoRD | 67.32 | 67 | 69.02 | 68.61 | 71.13 | 70.26 | 73.4 | 71.49 |
| CB | 42.86 | 44.64 | 46.43 | 42.86 | 48.21 | 44.64 | 42.86 | 37.5 |
| ANLI-R1 | 32.5 | 33.5 | 31.9 | 33.8 | 33 | 33.3 | 32.4 | 32.1 |
| ANLI-R2 | 30.3 | 34.7 | 30.5 | 34.6 | 30.6 | 33.1 | 31.5 | 33.5 |
| ANLI-R3 | 30.5 | 33.17 | 31.5 | 33.67 | 31.33 | 33.5 | 32.58 | 33.67 |
| Average | 41.96 | 42.71 | 42.84 | 43.85 | 44.85 | 44.51 | 46.21 | 45.13 |
| Avg over GEN Taks | 10.91 | 12.92 | 12.81 | 13.58 | 15.83 | 14.33 | 18.62 | 16.7 |
| Avg over RANK Tasks | 49.35 | 49.8 | 49.99 | 51.06 | 51.76 | 51.69 | 52.77 | 51.9 |
| Dev set log pplx | 2.90 | 2.923 | 2.856 | 2.867 | 2.79 | 2.81 | 2.718 | 2.76 |

Table 15: Downstream Eval numbers and development set log perplexity loss on 2.6B model size granularities.

| Downstream Task | Baseline-S | MatLM-S | Baseline-M | MatLM-M | Baseline-L | MatLM-L | Baseline-XL | MatLM-XL |
|---|---|---|---|---|---|---|---|---|
| TriviaQA (EM) | 18.58 | 18.64 | 19.83 | 21.41 | 25.17 | 24.9 | 28.84 | 28.01 |
| NaturalQuestions (EM) | 3.05 | 3.13 | 3.19 | 3.66 | 4.76 | 4.24 | 6.73 | 5.01 |
| WebQuestions (EM) | 5.61 | 6.74 | 4.43 | 6.3 | 6.1 | 6.74 | 8.27 | 7.78 |
| LAMBADA | 18.46 | 13.74 | 29.92 | 19.89 | 27.34 | 24.84 | 27.94 | 29.98 |
| HellaSwag | 46.41 | 46.01 | 49.04 | 48.94 | 52.87 | 52.2 | 57.14 | 55.33 |
| StoryCloze | 72.26 | 72.1 | 73.54 | 73.22 | 75.09 | 75.04 | 77.02 | 75.79 |
| WSC | 71.23 | 69.82 | 70.88 | 71.58 | 75.09 | 74.39 | 80 | 77.54 |
| WinoGrande | 56.83 | 57.85 | 57.62 | 56.91 | 60.93 | 59.19 | 62.19 | 59.59 |
| Winograd | 76.56 | 71.43 | 72.89 | 74.36 | 76.56 | 74.73 | 81.68 | 78.75 |
| SQuAD v2 (EM) | 34.89 | 37.97 | 34.33 | 40.07 | 34.89 | 42.24 | 43.47 | 42.59 |
| RACE-H | 33.62 | 34.76 | 35.59 | 35.85 | 36.91 | 36.82 | 38.91 | 37.28 |
| RACE-M | 47.63 | 47.49 | 49.44 | 49.51 | 50.77 | 50.07 | 53.34 | 51.67 |
| PIQA | 67.74 | 67.79 | 68.39 | 68.28 | 69.21 | 69.59 | 71.49 | 71.11 |
| ARC-C | 29.95 | 30.29 | 31.83 | 31.91 | 32.51 | 34.22 | 35.67 | 35.41 |
| ARC-E | 60.82 | 59.97 | 61.2 | 62.42 | 63.51 | 64.56 | 67.76 | 64.86 |
| OpenBookQA | 45.6 | 43.8 | 45.4 | 44.8 | 49 | 46.4 | 49 | 49.4 |
| BoolQ | 53.58 | 52.87 | 53.15 | 53.52 | 59.36 | 54.89 | 60.8 | 57.22 |
| COPA | 74 | 74 | 77 | 76 | 75 | 78 | 82 | 81 |
| RTE | 49.1 | 53.07 | 49.82 | 54.15 | 48.01 | 54.51 | 48.01 | 52.35 |
| WiC | 47.34 | 47.34 | 47.18 | 47.34 | 47.34 | 47.18 | 47.02 | 47.49 |
| MultiRC (F1) | 43.4 | 52.28 | 43.65 | 51.64 | 46.99 | 53.7 | 39.24 | 53.77 |
| ReCoRD | 71.34 | 71.9 | 72.79 | 72.97 | 74.86 | 74.57 | 76.71 | 75.32 |
| CB | 28.57 | 44.64 | 46.43 | 46.43 | 41.07 | 50 | 50 | 44.64 |
| ANLI-R1 | 32.4 | 32.3 | 30.4 | 32.3 | 32.5 | 32.1 | 31.2 | 31.5 |
| ANLI-R2 | 30.4 | 30.1 | 30.6 | 31 | 30.1 | 30.2 | 31.7 | 30.8 |
| ANLI-R3 | 30.75 | 30.83 | 31.25 | 31 | 33.5 | 30.92 | 32 | 31.92 |
| Average | 44.23 | 45.03 | 45.76 | 46.36 | 47.29 | 47.93 | 49.54 | 49.08 |
| Avg over GEN Taks | 16.12 | 16.04 | 18.34 | 18.26 | 19.66 | 20.59 | 23.05 | 22.68 |
| Avg over RANK Tasks | 50.93 | 51.94 | 52.29 | 53.05 | 53.86 | 54.44 | 55.85 | 55.37 |
| Dev set log pplx | 2.77 | 2.787 | 2.722 | 2.732 | 2.66 | 2.68 | 2.592 | 2.63 |

Table 16: Downstream eval numbers and development set log perplexity on 2.6B MatLM Mix'n'Match granularities. For original granularities, please refer to Table 15. First row represents the non-embedding parameters of the model.

| Downstream Task | 830M | 1B | 1.11B | 1.32B | 1.43B | 1.55B | 1.65B |
|---|---|---|---|---|---|---|---|
| TriviaQA (EM) | 18.89 | 22.43 | 23.8 | 25.77 | 26.26 | 26.15 | 26.6 |
| NaturalQuestions (EM) | 3.49 | 3.77 | 4.02 | 4.07 | 4.46 | 4.65 | 5.12 |
| WebQuestions (EM) | 5.95 | 6.1 | 6.64 | 6.69 | 6.94 | 6.69 | 6.69 |
| LAMBADA | 16.34 | 20.16 | 23.07 | 24.8 | 24.32 | 25.87 | 29.13 |
| HellaSwag | 47.98 | 50.46 | 51.29 | 52.78 | 53.75 | 54.16 | 54.56 |
| StoryCloze | 73.01 | 73.33 | 74.83 | 75.2 | 75.68 | 75.41 | 75.63 |
| WSC | 70.88 | 70.53 | 74.04 | 72.98 | 74.74 | 73.33 | 77.19 |
| WinoGrande | 57.85 | 58.88 | 60.93 | 58.88 | 59.67 | 60.06 | 59.91 |
| Winograd | 73.26 | 73.26 | 76.19 | 74.36 | 76.56 | 77.66 | 78.02 |
| SQuAD v2 (EM) | 36.49 | 39.72 | 38.05 | 41.33 | 41.08 | 40.26 | 41.36 |
| RACE-H | 34.71 | 35.93 | 35.48 | 36.74 | 36.62 | 36.22 | 36.96 |
| RACE-M | 46.59 | 48.89 | 49.44 | 50.28 | 50.42 | 51.32 | 50.91 |
| PIQA | 68.5 | 69.04 | 69.53 | 70.4 | 70.46 | 70.51 | 70.29 |
| ARC-C | 31.06 | 33.11 | 33.19 | 34.81 | 35.75 | 35.84 | 34.56 |
| ARC-E | 62.29 | 62.58 | 62.63 | 64.86 | 65.99 | 65.49 | 64.69 |
| OpenBookQA | 44.6 | 46.2 | 46.8 | 47 | 47.4 | 47.4 | 47.6 |
| BoolQ | 54.86 | 55.08 | 54.46 | 55.78 | 58.38 | 57.19 | 56.88 |
| COPA | 76 | 76 | 75 | 80 | 77 | 80 | 80 |
| RTE | 53.43 | 53.79 | 53.79 | 52.71 | 53.79 | 54.51 | 53.79 |
| WiC | 47.34 | 47.34 | 47.18 | 47.34 | 47.18 | 47.34 | 48.12 |
| MultiRC (F1) | 53.34 | 53.85 | 52.97 | 54.23 | 57.57 | 55.09 | 54.91 |
| ReCoRD | 72.21 | 73.25 | 73.98 | 74.43 | 74.72 | 75.05 | 75.37 |
| CB | 48.21 | 46.43 | 48.21 | 50 | 50 | 44.64 | 55.36 |
| ANLI-R1 | 32.4 | 32.1 | 32 | 32.4 | 32.3 | 31.4 | 32.4 |
| ANLI-R2 | 30.5 | 30.6 | 30.6 | 30.6 | 30.7 | 30.4 | 31.4 |
| ANLI-R3 | 31.17 | 31.17 | 31.17 | 31.5 | 31 | 31.5 | 31.33 |
| Average | 45.82 | 46.69 | 47.28 | 48.07 | 48.57 | 48.39 | 49.18 |
| Avg over GEN Taks | 16.23 | 18.44 | 19.12 | 20.53 | 20.61 | 20.72 | 21.78 |
| Avg over RANK Tasks | 52.87 | 53.42 | 53.99 | 54.63 | 55.22 | 54.98 | 55.71 |
| Dev set log pplx | 2.774 | 2.729 | 2.706 | 2.68 | 2.675 | 2.663 | 2.65 |

