# OpenReview forum: "MatFormer: Nested Transformer for Elastic Inference"
_NeurIPS.cc/2023/Workshop/WANT — WANT@NeurIPS 2023 Oral_

### Official Review · Reviewer_DDgC · 2023-10-23
**The paper proposes to jointly train a series of models of varying model size but ends up training models of the same size with different low rank for the feed forward layer**

**Confidence:** 4

**Review:**

Summary Of The Paper:
The paper gives a method for jointly training models of different sizes instead of training each model independently. In the paper, the author specifically varies the hidden layer sizes of the feed forward layer and shows consistency of performance across the various sub models against the baseline of training each model independently.

Strength And Weaknesses:

Strengths: The paper evaluates the proposed approach over various domains such as Language Modeling and Vision. The paper also goes over using the sub model in speculative decoding and the additional speedup over independently training the smaller model.

Weakness: The paper claims that training their joint model is the same as being able to get a series of models with different parameter counts from training a single model instead of training many different model configurations independently like in LLaMa or Pythia. However, each of the submodels they get from this training procedure is actually a large model with a low rank in its hidden layer for the feed forward module. That is, each sub model in MatFormer still has the same dimension size as the large model. While, for example, Pythia 90m has a hidden size of 512 and Pythia 12B has a hidden size of 5120. This is why the smallest sub model has 1.5B parameter count while the whole model has 2.6B. So the submodel is around 60% the size of the whole model. The training time improvement shown by the authors for the joint model is 15% lower than training each of the sub models independently. But as stated before if the sub models were actually smaller training time for training them independently would probably have been lower than doing this joint training.

Clarity, Quality, Originality And Significance:

Clarity: The paper presents the idea clearly and is easy to read.

Quality: The paper tries to tackle an important problem but probably is not solving it.

Originality: The approach to getting a series of models of varying size by training just a single model is interesting.

Significance: The increase in performance for speculative decoding by using a series of models trained jointly could potentially be used to reduce serving costs.

---

### Official Review · Reviewer_3L7J · 2023-10-23
**Interesting idea that can bring a twist to the LLM world**

**Confidence:** 4

**Review:**

The authors follow a very interesting idea: bring the idea of matryoshka training to LLMs. In Matryoshka training, several models with shared weights of different sizes are trained at the same time. This is possible if the models share the same structure and only vary in dimensionality. For LLMs, the authors show that it is possible to choose a model with the desired number of parameters after the actual model training.

As a practitioner in LLM pretraining, I find this idea very exciting and I believe can bring an innovative push to the field. At present, I'm not convinced the method is ready for large-scale rollout, but I'm very happy with the originality of this idea.

The paper is well written and understandable, and the chosen benchmarks are justified. To me, it appears that not much was missing for a full NeurIPS paper. For the WANT workshop, the paper can definitely be accepted.

---

### Official Review · Reviewer_3163 · 2023-10-24
**The authors introdusing a novel and agile general-purpose architecture, named MatFormer, which is extension of original Transformer idea. Key features are scalability, adaptiveness and relative easy implementation. Nested structure(matryoushka-like) allows to sacrifice a part of architecture without additional trainings and dramatic performance drops(0.5% in order to gain 40% less overhead). Paper also numericaly proves the effictiveness of the proposed architecture via comparison with standard Transformer's scalability. Constant trend among different models' scales makes it fair to claim two models similar in terms of performance increase/degradation.**

**Confidence:** 4

**Review:**

**Strengths**:
- Qualitative review of approaches dealing with overhead of training and deploying Large Language Models(**LLMs**), including *speculative decoding*
- Wide range of problem statements are covered, including ablation study of sub-models of different parameter count for both proposed architectures(MatFormer, MatViT) and classic ones(NLP Transformer and Visual Transformer, **ViT**), Computer VIsion(**CV**) and Natural Language Processing(**NLP**) applications
- Comparing more than one dataset and averaging the results helps to detect general trends of metrics
- Exhaustive number of plots, figures, tables are presented in appendix makes it easiler to get deep in details
- Clear story-telling

**Weaknesses**:
- Table 7: missing annotation for what metric is used to compare relative performance on ViT vs MatViT
- Missing framework details
- I find Table 6 disputable and lacking motivation for these parameter reciprocal ratio
- Honestly, Figure 1 doen't really help with the understanding of inter-connection of nested Transformer blocks

**Recommendation for improvement**:
- Bold markdown style helps to perceive significant number that you wanted to highlight, while describing your results in tables
- Table 10 annotation intersects with table view
- line 543 typo "Baselime"

---

### Official Review · Reviewer_7yw4 · 2023-10-25
**Application of Matryoshka representation learning to transformers**

**Confidence:** 4

**Review:**

The FFN block in a transformer has a single hidden layer so it is possible to modulate the size of that hidden layer without otherwise affecting the network. This paper applie Matryoshka Representation Learning (MRL) to that block, the clearest explanation of which is the following pseudocode ([based on the code in Appendix A of the MRL paper](https://homes.cs.washington.edu/~kusupati/pubs/kusupati22.pdf)):
```
class MRL_FFN_Layer(nn.Module):
    def __init__(self, nesting_list: List, num_classes=1000, **kwargs):
        super(MRL_Linear_Layer, self).__init__()
        self.nesting_list=nesting_list # list of nested hidden layer sizes [<num_feat>, ...]
        self.num_classes=num_classes
        self.nesting_classifier_0 = nn.Linear(self.nesting_list[-1], self.num_classes, **kwargs))
        self.m = 0 # index of the nesting list the layer is currently set to

    def set_m(self, m):
        self.m = m

    def forward(self, x):
        num_feat = self.nesting_list[self.m]
        return torch.matmul(
            x[:, :num_feat],
            (self.nesting_classifier_0.weight[:, :num_feat]).t()
        )
```
The authors suggest this will be useful to make the downstream model a more flexible resource because smaller and larger models may be readily extracted from the same set of weights. They further describe a method of routing at inference time through the differently sized blocks from different model sizes (imagine, using the above linear block with different blocks set to different `m` values). The authors name this "mix-n-match" and it allows interpolation between model scales.

The experiments demonstrate the claims that this method is able to train functional models, able to match the performance of a baseline transformer model up to 2.7B parameters. In addition the authors explore their thesis on the importance of elastic models and demonstrate  improved speculative decoding and interpolated model scales in between their smallest and largest models.

Experiments are also performed on Vision transformers performing image classification with similar positive results. Training the network in this way does not adversely affect the performance of the model collection. The authors then demonstrate the value of an elastic model by demonstrating strong performance in adaptive image retrieval.

However, the experiments do lack a strong comparison method, any similar proposed method from the literature to provide an elastic model. I can think of two potential candidates:

* Traditional dropout on the FFN hidden layer
* [Nested Dropout](https://proceedings.mlr.press/v32/rippel14.html) on the FFN hidden layer

Then it is possible to generate networks in the same size ranges as the networks described in this paper by randomly sampling them according to the dropout probabilities.

Pros:

- Clear and well written
- "Mix-n-match" finding that various paths through the network create functioning models is surprising
- Comprehensive experiments validating the elastic inference claims
- Solves the problem as stated

Cons:

- Requires independent execution of each model size during training
- Deviation from a traditional transformer design is small and this explains why the performance is close to baseline
- Range of model scales available is limited by proportion of parameters in the FFN block, significant parameters are in the QKV linear layers in a transformer for example
- No comparison to competing methods, see above

---

### Meta-Review · Area_Chair_zWwm · 2023-10-27

**Recommendation:** Accept (Oral)
**Confidence:** 4

**Metareview:**

Here is a brief summary of the paper’s strengths and weaknesses based on submitted reviews:

**Strengths:**
* Most reviewers found the paper to be well-written, with clear explanations of the key concepts.
* Reviewers agree that the paper targets an important and relevant problem.
* Strong evaluation with results on datasets and networks from varying domains such as computer vision and NLP.
* Surprising accuracy results with mix n’ match.


**Weaknesses:**
* Lack of comparisons to similar related work. For example, traditional and nested layer dropout.
* Multiple reviewers pointed out the potentially costly training process.
* Limited novelty beyond the traditional Transformer architecture and existing work on Matryoshka Representation Learning.

The overall sentiment for the paper based on the submitted reviews appears to be positive. I recommend acceptance (oral).

---

### Decision · Program_Chairs · 2023-10-28

**Decision:**

Accept (Oral)

**Comment:**

We thank the authors for their time and contribution to WANT and we are pleased to share that after the reviewing process the paper has been accepted. Congratulations! We encourage the authors to consider reviewers' feedback for the improvement of the camera-ready version. We hope to see you in person at the workshop and brainstorm on efficient training research together!